# An E2-E3 pair contributes to seed size control in grain crops

Sha Tang [1,6], Zhiying Zhao[1,6], Xiaotong Liu[2,3,6], Yi Sui[1], Dandan Zhang[1], Hui Zhi[1], Yuanzhu Gao[1], Hui Zhang[1], Linlin Zhang[4], Yannan Wang[1], Meicheng Zhao[3], Dongdong Li[5], Ke Wang [1], Qiang He [1], Renliang Zhang[1], Wei Zhang[1], Guanqing Jia[1], Wenqiang Tang[2], Xingguo Ye [1], Chuanyin Wu[1] & Xianmin Diao [1] ✉

Understanding the molecular mechanisms that regulate grain yield is important for improving agricultural productivity. Protein ubiquitination controls various aspects of plant growth but lacks understanding on how E2-E3 enzyme pairs impact grain yield in major crops. Here, we identified a RING-type E3 ligase SGD1 and its E2 partner SiUBC32 responsible for grain yield control in *Setaria italica*. The conserved role of *SGD1* was observed in wheat, maize, and rice. Furthermore, SGD1 ubiquitinates the brassinosteroid receptor BRI1, stabilizing it and promoting plant growth. Overexpression of an elite *SGD1* haplotype improved grain yield by about 12.8% per plant, and promote complex biological processes such as protein processing in endoplasmic reticulum, stress responses, photosystem stabilization, and nitrogen metabolism. Our research not only identifies the SiUBC32-SGD1-BRI1 genetic module that contributes to grain yield improvement but also provides a strategy for exploring key genes controlling important traits in Poaceae crops using the Setaria model system.

With the continuous increase in global population, the effects of climate change, and the decreasing availability of arable land, global food security is becoming increasingly challenging[1]. Thus, improving crop yield to feed more people has always been the primary goal of agriculture research. For staple crops in the Poaceae family, such as maize (*Zea mays*), rice (*Oryza sativa*), and wheat (*Triticum aestivum*), productivity is determined by grain yield, which consists of several key components, including panicle (or cob) number, grain number per panicle, and 1000-grain weight. In addition, grain size is an important component of grain yield[2]. Foxtail millet (*Setaria italica*), which also belongs to the Poaceae family, is an important cereal crop in East Asia and has recently been proposed as an ideal model system, along with

its wild ancestor *Setaria viridis*, due to its morphological and genetic characteristics[3,4]. The inflorescence (panicle) structure of foxtail millet is similar to that of other major crops in the Poaceae family, such as maize, rice, wheat, sorghum, and barley. Investigation on *Sparse panicle1* is one of the examples that extend knowledge on inflorescence architecture from *Setaria* to maize[5]. Therefore, *Setaria* is a good model for grain yield-related research for the Poaceae family.

Grain yield is a complex quantitative trait that is determined by various loci and genetic pathways in plants[2]. The ubiquitination cascade pathway involves ubiquitin-activating (E1), ubiquitin-conjugating (E2), and ubiquitin ligase (E3) enzymes[6], and is fundamental in controlling grain yield by regulating proteasomal degradation, protein

[1]Institute of Crop Sciences, Chinese Academy of Agricultural Sciences, Beijing 100081, China. [2]Key Laboratory of Molecular and Cellular Biology of Ministry of Education, Hebei Collaboration Innovation Center for Cell Signaling, Hebei Key Laboratory of Molecular and Cellular Biology, College of Life Sciences, Hebei Normal University, Shijiazhuang 050024, China. [3]Key Laboratory of Agricultural Water Resources, Hebei Laboratory of Agricultural, Water-Saving, Center for Agricultural Resources Research, Institute of Genetics and Developmental Biology, Chinese Academy of Sciences, Shijiazhuang 050021, China. [4]College of Biological Sciences, China Agricultural University, Beijing 100193, China. [5]College of Agronomy and Biotechnology, China Agricultural University, Beijing 100193, China. [6]These authors contributed equally: Sha Tang, Zhiying Zhao, Xiaotong Liu. ✉e-mail: diaoxianmin@caas.cn

stability, and localization[2]. Several ubiquitin-related proteins involved in grain yield control have been identified, such as DA1, DAR1, and DA2 in *Arabidopsis*[7,8] and GW2 in rice[9].

Brassinosteroid (BR) is a phytohormone that broadly regulates plant growth and development, as well as stress responses[10]. BRASSINOSTEROID INSENSITIVE1 (BRI1) is the key BR receptor that can bind BR through its ectodomain[11]. Rice BRI1 loss-function mutant *d61* exhibited BR-insensitive phenotypes, including dwarf culms, erect leaves, low leaf angle, and small grains[12,13]. Several E3 ubiquitin ligases are involved in BR signaling, including *SINAT*[14], *KIB1*[15], *ELF1*[16], *TUD1*[17], *PUB12*, and *PUB13*[18]. Among them, the plant U-box E3 ubiquitin ligase PUB12 and PUB13 can interact with and ubiquitinate BRI1, resulting in BRI1 internalization[18]. The ubiquitination of BRI1 in *pub12/13* mutants was significantly reduced compared to wild-type plants, resulting in an accumulation of BRI1 proteins. However, BRI1 internalization and degradation were still observed in *pub12/13* mutants, and the double mutant had a little alteration in BR sensitivity[18]. These results indicated that, in addition to PUB12/13, there may be other E3 ligase(s) that regulate either the protein stability, activity, or localization of the BRI1 receptor through ubiquitination modification, which still needs to be explored. In foxtail millet, our group identified an LRR receptor-like kinase, DROOPY LEAF1 (DPY1), which orchestrates early BR signaling and regulates leaf droopiness[19].

Here, we employed a Setaria model system to investigate grain yield control in Poaceae crops. Two foxtail millet mutants were isolated, which showed reduced grain yield and semi-dwarf phenotypes. Positional cloning revealed that the causal gene *SGD1* (*small grain and dwarf1*) encodes a RING-type E3 ubiquitin ligase. Phylogenetic analysis suggests that *SGD1* is orthologous to *TT3.1* (Thermo-tolerance 3.1)[20], a key regulator of the response to heat stress in rice. However, the molecular mechanisms of *SGD1* in grain yield control remain to be elucidated. Our study found that SGD1 interacts and acts genetically with the E2 ubiquitin-conjugating enzyme SiUBC32 in regulating grain yields in Poaceae crops. Furthermore, SGD1 can directly interact with and ubiquitinate the BR receptor SiBRI1, which stabilizes BR-signaling and finally enhances grain yields. Comprehensive genetic complementation tests and transgenic experiments support that the "SiUBC32-SGD1-SiBRI1" genetic module works in grain yield regulation in Poaceae crops, and could be used for high-yield breeding.

## Results

### Identification of two mutant alleles related to grain yield in *Setaria italica*

Plant density, grain number per panicle, and 1000-grain weight determine grain yield in foxtail millet. Our group identified various foxtail millet mutants related to panicle and seed development, including *SiAUX1*[21] and *SiBOR1*[22]. Among them, two mutant alleles associated with small grain and dwarf phenotypes (*sgd1-1* and *sgd1-2*), were of particular interest. Both *sgd1-1* and *sgd1-2* mutants exhibited reduced plant height (Fig. 1a–c and Supplementary Table 1), smaller panicles and grains (Fig. 1d–f), fewer grain numbers, and lower 1000-grain weight (Fig. 1g) than those of wild-type (WT) Yugu1, resulting in severe grain yield losses (Fig. 1g, h and Supplementary Table 1). F₁ plants of the *sgd1-1* × *sgd1-2* cross were more similar to either mutant than to the WT, suggesting that the two mutations are allelic (Fig. 1f and Supplementary Fig. 1). To investigate if a smaller seed size in mutants was caused by changes in cell size or cell number, we employed resin sections and scanning electron microscope (SEM) analyses. Cell length and width were decreased in mutants, while cell number was similar between WT and mutants, suggesting that decreased seed size was due to reduced cell expansion in mutants (Fig. 1i–k).

### *SGD1* encoding a RING-type E3 ubiquitin ligase is crucial for grain yields regulation in foxtail millet

Through genetic analysis, map-based cloning, and Mutmap analysis, we identified effective homozygous mutations in the coding region of the candidate gene, *Seita.9G123200*, in *sgd1-1* and *sgd1-2* (Fig. 2a and Supplementary Tables 2, 3). These results were confirmed by Sanger sequencing of genomic DNA and transcripts of the two mutants (Supplementary Fig. 1). In *sgd1-1*, A to T transversion occurred at the splice acceptor of the 8th exon, leading to intron retention and translation termination. In *sgd1-2*, C to T transition was identified on the 13th exon, producing an early stop codon (CGA to TGA, Fig. 2a and Supplementary Fig. 1).

To confirm that *Seita.9G123200* is responsible for the mutant phenotype, CRISPR/Cas9 genome editing, and transgenic functional complementation were performed in foxtail millet. Two independent positive transgenic lines (*CR-sgd1*-L1 and *CR-sgd1*-L2) without *Seita.9G123200* phenocopied *sgd1-1* and *sgd1-2*, exhibiting reduced plant height, shorter panicles, and smaller grains compared to that of the wildtype (Fig. 2b–d and Supplementary Fig. 2). These mutant phenotypes were fully rescued by expressing the WT *Seita.9G123200* (*SGD1*) under control of the native promoter (*pSGD1::SGD1-GFP*) in a mutant background (Fig. 2b–f and Supplementary Fig. 2). SEM analysis showed that the changes in grain size in transgenic lines were also associated with variations in cell size rather than changes in cell number (Fig. 2e, f). Collectively, we demonstrate that *Seita.9G123200* is responsible for *SGD1*.

Subcellular localization assays were carried out using SGD1-GFP fusion protein and FM4-64 as a plasma membrane (PM) marker. We found that SGD1-GFP signals merged well with FM4-64 (Fig. 2g, h), suggesting that SGD1 is located at the PM, consistent with previous findings in the rice ortholog TT3.1[20]. However, we also observed different GFP signals that did not colocalize with the PM marker at certain subcellular organelles, probably the endoplasmic reticulum (ER), as indicated by the yellow arrows (Supplementary Fig. 3a). We then co-expressed SGD1-GFP and the ER marker HDEL-mCherry in *Nicotiana benthamiana* cells. The results showed that SGD1 colocalized with the ER marker (Fig. 2i, j), suggesting that SGD1 has a molecular function in the ER in addition to its previously reported function as a thermo-sensor at the PM[20].

The expression pattern of *SGD1* in foxtail millet was analyzed using multi-organs RNA-seq datasets (Supplementary Data 1) and *SGD1*-promoter-driven *GUS* reporter (*pSGD1::GUS*) transgenic plants. The expression of *SGD1* was detected in roots, elongating stem, leaves, and flowering panicles, with relatively higher expression in young panicles, milking seeds, and seedling stage roots (Fig. 2k and Supplementary Data 1). Histochemical staining showed positive GUS signals in leaves, roots, young panicles, bristles, and developing flowers (Supplementary Fig. 3b), supporting the role of *SGD1* in regulating panicle and seed development.

### SGD1 functions conservatively in major crops of the Poaceae family

A genome-wide analysis identified 495 RING-type proteins in foxtail millet (Supplementary Data 2). Bioinformatic analysis indicated that the *SGD1* gene encodes a RING-type E3 ubiquitin ligase, which contains a seven tandem transmembrane Fragile-X-F domain and a C3HC4 RING-type zinc finger domain (Fig. 2a). Phylogenetic analysis suggests that *SGD1* is orthologous to *TT3.1* (Supplementary Fig. 4), which was reported to be a positive regulator in promoting rice thermotolerance[20]. In addition, three orthologs of *SGD1* were identified in *Arabidopsis* based on the phylogenetic tree (Supplementary Fig. 4) and BLAST identity (Supplementary Data 3), among which *PPRT1* (*AT1G68820*) was associated with salt stress response[23]. We knocked out *OsSGD1* in rice (Fig. 3a). As expected, two CRISPR-edited rice *Ossgd1* lines showed shorter panicles, smaller grains, and fewer grain

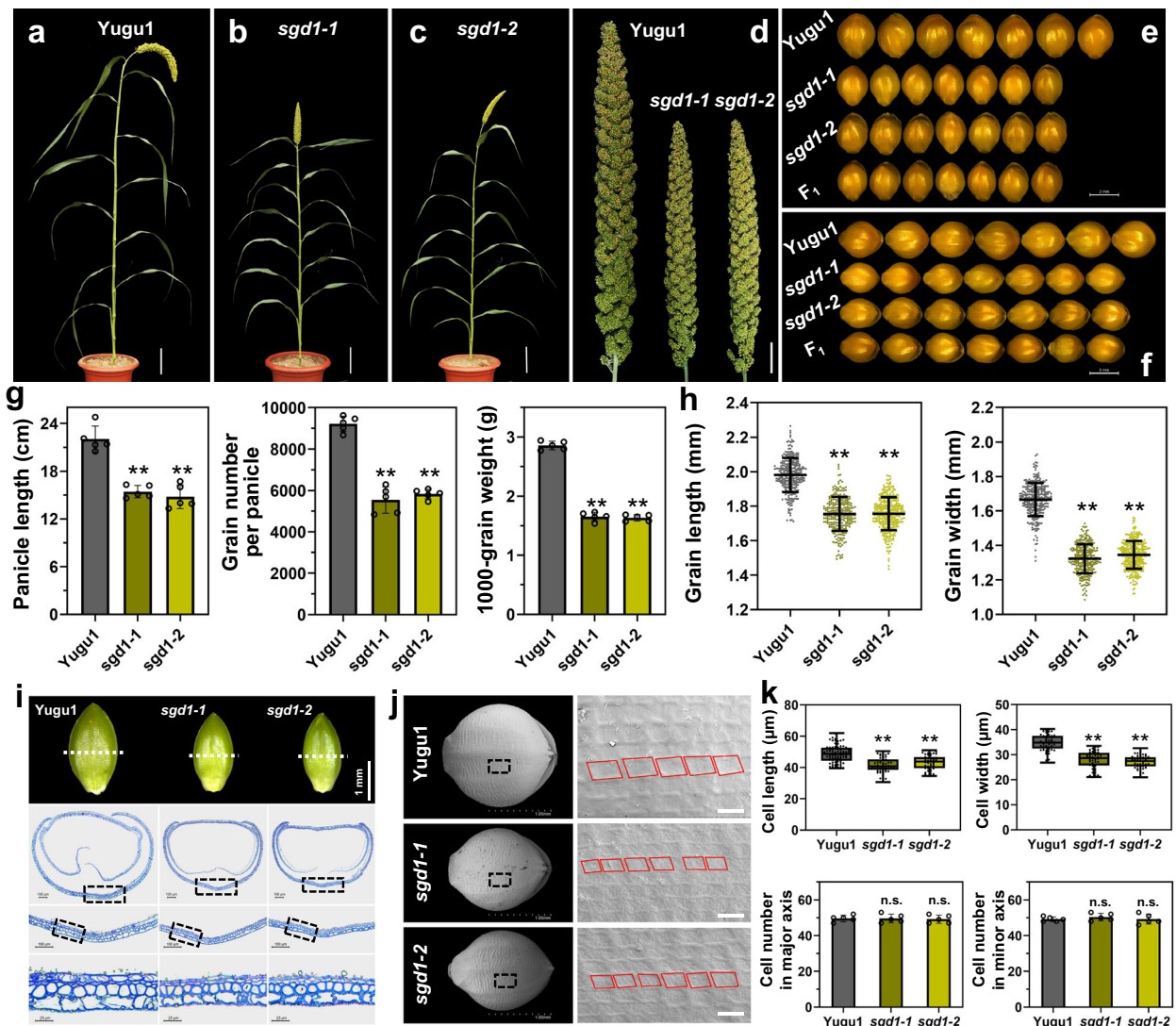

**Fig. 1 | Phenotypic characterization of foxtail millet *sgd1-1* and *sgd1-2* mutants.**
**a–c** Wild-type Yugu1 (**a**), mutant *sgd1-1* (**b**), and mutant *sgd1-2* (**c**) grown in the field at the grain-filling stage. Bar = 10 cm. **d** Panicles of Yugu1, *sgd1-1*, and *sgd1-2* plants. Bar = 2 cm. **e, f** Grain width (**e**) and grain length (**f**) of Yugu1, *sgd1-1*, *sgd1-2*, and F₁ (*sgd1-1* × *sgd1-2*) plants. Bar = 2 mm. **g** Panicle length, grain number per panicle, and 1000-grain weight in Yugu1, *sgd1-1*, and *sgd1-2* plants. Five biological replicates were used for each measurement (*n* = 5). Error bars indicate mean ± SD. Significant differences between wild-type and mutant plants were determined by unpaired two-sided Student's *t*-tests, (*$P < 0.01$, **$P < 0.001$). **h** Grain length and grain width of Yugu1, *sgd1-1*, and *sgd1-2* plants (*n* > 100, unpaired two-sided Student's *t*-tests, *$P < 0.01$, **$P < 0.001$. Error bars indicate mean ± SD). **i** Resin sections of immature grains of Yugu1, *sgd1-1*, and *sgd1-2* plants (from left to right). The white dashed line represents the section position. The black box represents

the magnification position. **j** Scanning electron microscopy (SEM) analysis of lemmas of Yugu1, *sgd1-1*, and *sgd1-2* plants. The black box represents the magnification position with an enlarged view on the right. The red box indicates the cell size. Bar = 1 mm (left), 50 μm (right). These experiments in (**i, j**) were repeated five times independently with similar results. **k** Cell length, cell width, and cell number in the major and minor axes in mature seeds of Yugu1, *sgd1-1*, and *sgd1-2* plants. Cell length and width were measured in more than 50 cells. Cells from five seeds were counted. The two ends of the box plot and the upper, middle, and lower box lines represent the upper edge, lower edge, median, and two quartiles of values in each group. Error bars indicate mean ± SD. Significant differences were determined by unpaired two-sided Student's *t*-tests. *$P < 0.01$, **$P < 0.001$ vs. Yugu1 plants. n.s. means not statistically significant. Source data are provided as a Source Data file.

numbers per panicle (Fig. 3b–d and Supplementary Fig. 5). To test whether the role of *SGD1* in regulating grain yield was conserved across Poaceae species, we isolated two *SGD1* homologs in maize (*Zm00001d013466* and *Zm00001d033674*) (Supplementary Fig. 4). Due to its higher protein sequence identity to SGD1 (Supplementary Data 3), we introduced *Zm00001d013466* (*ZmSGD1*) under control of the maize *Ubiquitin* promoter (*Ubi::ZmSGD1-GFP*) into a foxtail millet *sgd1* mutant background (*CR-sgd1*-L1). The mutant phenotypes were also rescued by *ZmSGD1* (Fig. 2b–f). Hexaploid wheat is one of the most important crops worldwide. Three copies of the *SGD1* gene were identified in wheat A, B, and D genomes (*TaSGD1A*, *TaSGD1B*, and *TaSGD1D*) (Supplementary Fig. 4). We successfully created a null

*Tasgd1* triple mutant (*Ta-sgd1a/1b/1d*) in wheat using CRISPR/Cas9 genome editing (Fig. 3e–g). Notably, null *sgd1* mutants in wheat also exhibited significant reductions in panicle length, grain number per panicle, grain size, and 100-grain weight (Fig. 3f–h). In addition, the awn length of the *Tasgd1* triple mutant was shorter than that of WT plants (Supplementary Fig. 5). In summary, the results of transgenic experiments suggest that *SGD1* has a conserved function in regulating grain yield across the Poaceae family.

**SiUBC32 directly interacts with SGD1 and regulates grain yield**
To further investigate the molecular functions of SGD1, we utilized split-ubiquitin membrane-based yeast two-hybrid (Y2H) screening to

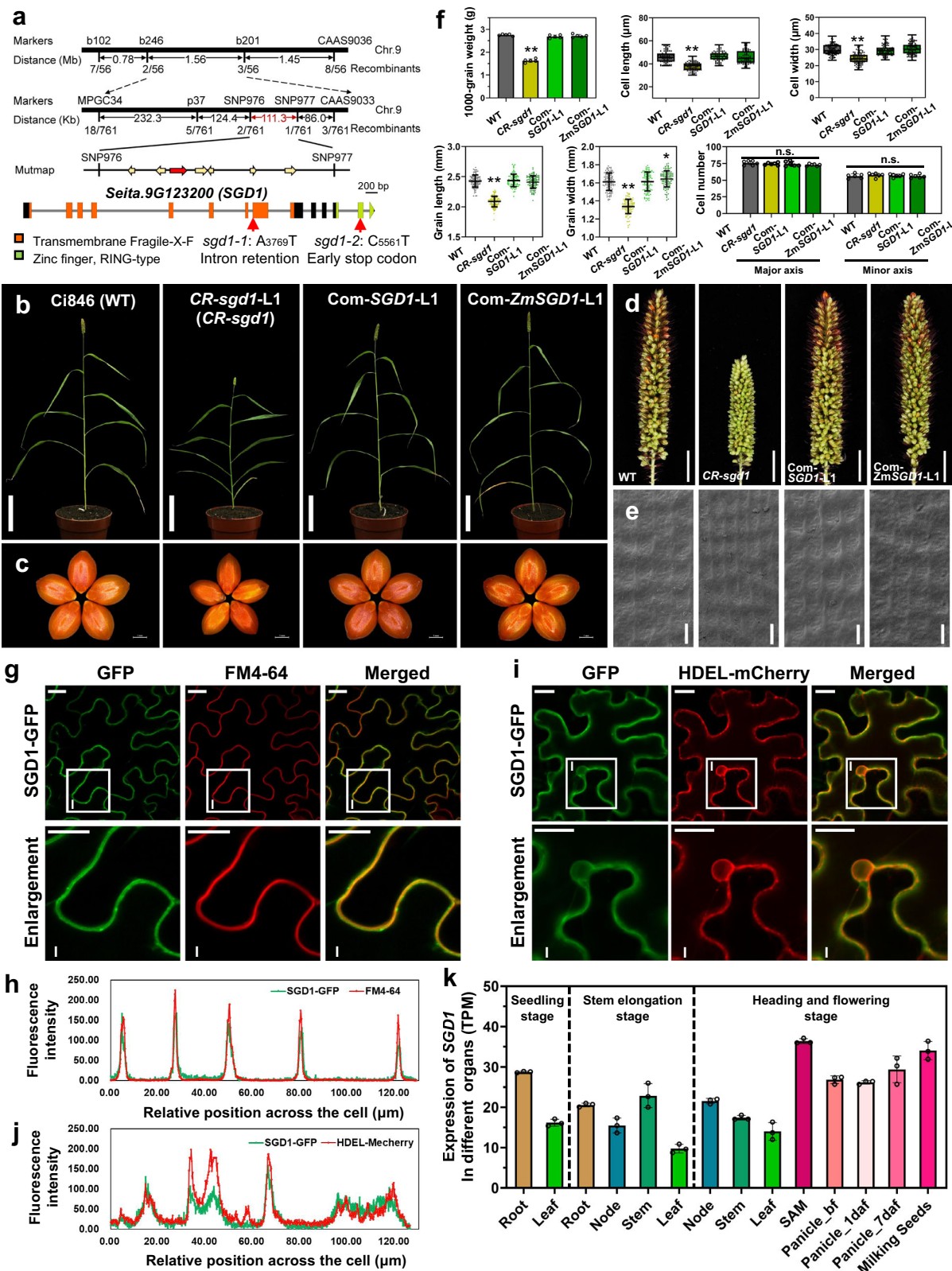

identify SGD1-interacting proteins. A few putative candidates were identified, including *Seita.9G428900*, an ortholog of *UBC32* (*AT3G17000*, Supplementary Fig. 6), which is a ubiquitin-conjugating enzyme (E2) in *Arabidopsis* (Supplementary Data 4). Since the E2-E3 complex is essential for the ubiquitination of substrate proteins, we chose *Seita.9G428900* (*SiUBC32*) for further studies. In *Arabidopsis*, UBC32 localizes to the ER membrane, improving stress tolerance and

BR-mediated growth[24]. In line with the previous study, our results showed that SiUBC32 and SGD1 colocalized in the ER in both *S. italica* protoplasts and *N. benthamiana* leaf cells (Fig. 4a and Supplementary Fig. 7a) and had similar spatiotemporal expression patterns in various *S. italica* organs (Supplementary Fig. 7b). These results suggest that SiUBC32 partners with SGD1. To confirm the SGD1–SiUBC32 interaction, we carried out Y2H assays using SGD1 as bait and SiUBC32 as prey

**Fig. 2 | Positional cloning and functional characterization of *SGD1*. a** Map-based cloning of *SGD1*. The gene structure of *SGD1* and mutation sites in *sgd1-1* and *sgd1-2* are indicated. **b** Wild-type (WT) Ci846, *SGD1* knockout plants (*CR-sgd1*-L1), and rescued plants (Com-*SGD1*-L1 and Com-*ZmSGD1*-L1) at the grain-filling stage. All plants were grown in a growth chamber under a 10-h light/14-h dark cycle at 30/26 °C. Bar = 10 cm. **c** Mature seeds of WT, *CR-sgd1*-L1, Com-*SGD1*-L1, and Com-*ZmSGD1*-L1 plants (from left to right). Bar = 1 mm. **d** Panicles of WT, *CR-sgd1*-L1, Com-*SGD1*-L1, and Com-*ZmSGD1*-L1 plants. Bar = 1 cm. **e** SEM analysis of the lemma of WT, *CR-sgd1*-L1, Com-*SGD1*-L1, and Com-*ZmSGD1*-L1 plants (from left to right). Bar = 30 μm. This experiment was repeated five times independently with similar results. **f** Thousand-grain weight (*n* = 5), grain length and width (*n* > 100), cell length and width (*n* > 50), and cell number (*n* = 5) in WT, *CR-sgd1*-L1, Com-*SGD1*-L1, and Com-*ZmSGD1*-L1 plants. The two ends of the box plot and the upper, middle, and lower box lines represent the upper edge, lower edge, median, and two quartiles of

values. Data were means ± SD. Significant differences were determined using unpaired two-sided Student's *t*-tests. **P < 0.001 vs. WT plants. n.s. means not statistically significant. **g** SGD1-GFP colocalized with the membrane probe FM4-64. The white box marked with I represents the magnification position with an enlarged view. Bar = 20 μm. **h** Fluorescence intensity of SGD1-GFP and FM4-64 signals across the cell. **i** SGD1-GFP colocalized with the ER marker HDEL-mCherry in *Nicotiana benthamiana* leaf cells, along with a magnified view of the boxed areas. Bar = 20 μm. These experiments in (**g**, **i**) were repeated three times independently with similar results. **j** Fluorescence intensity of SGD1-GFP and HDEL-mCherry signals across the cell. **k** Expression of *SGD1* in foxtail millet organs at different growth stages. Error bars indicate mean ± SD. *n* = 3 biological replicates. The gene expression profile of different organs of Yugu1 plants is shown in Supplementary Data 1. Source data are provided as a Source Data file.

and in vitro GST pull-down assays (Fig. 4b, c). The results support that SGD1 interacts with SiUBC32 in vitro. To verify this association in vivo, we performed a split luciferase complementation assay (LCA) in *N. benthamiana* leaves. Luciferase activity occurred only when we co-expressed SGD1-nLUC (SGD1 fused to the N-terminus of LUC) and SiUBC32-cLUC (SiUBC32 fused to the C-terminus of LUC) (Fig. 4d). In vivo co-immunoprecipitation (Co-IP) assay was also performed in *S. italica* leaf protoplasts co-transfected with a construct encoding SGD1-HA and a construct encoding SiUBC32-FLAG. We found that SGD1 coimmunoprecipitated with SiUBC32 (Fig. 4e). These results demonstrate that SGD1 interacts with SiUBC32.

Self-ubiquitination is critical for the E2-E3 interaction. Thus, we investigated the ubiquitin-conjugating enzyme activity of SiUBC32 and the ubiquitin ligase activity of SGD1 using a bacterial ubiquitination system[25]. Recombinant SiUBC32-S, E2CK-S (an E2-conjugating enzyme used as a negative control), SGD1c-Myc (truncated SGD1 lacking the transmembrane domain as depicted in Supplementary Fig. 7c, d), ubiquitin-FLAG, and UBA1(E1)-S proteins were expressed in *E. coli* for ubiquitination assays. Ubiquitin conjugation was observed for SiUBC32-S (Fig. 4f). Furthermore, a high molecular weight smear ladder was detected with SGD1c-Myc co-expressed with E1, SiUBC32, and ubiquitin, but not with E2CK, suggesting that SiUBC32-SGD1 was a functional E2-E3 ubiquitin enzyme pairing partner. To assess whether the RING-type zinc finger domain of SGD1 is essential for its E3 ligase activity, we mutated cysteine 426 to alanine (C426A) and histidine 443 to alanine (H443A) in the domain to form mSGD1c-Myc (catalytically inactive) (Supplementary Fig. 7c, d). No ubiquitin signal was detected with mSGD1c recombinant proteins in the presence of ubiquitin, UBA1, and SiUBC32 (Supplementary Fig. 7e), indicating that SGD1 ligase activity is dependent on the RING domain.

To determine the genetic relationship between SiUBC32 and SGD1 in regulating grain yield, we created a foxtail millet *Siubc32* single mutant and a *Siubc32/sgd1* double mutant using CRISPR/Cas9 genome editing. Two independent *Siubc32* mutants (*CR-Siubc32*-L1 and *CR-Siubc32*-L2) containing loss-of-function mutations were obtained (Supplementary Fig. 8). *Siubc32* mutants presented a dwarf phenotype, shorter panicles, and smaller grain size than WT Ci846, which mimicked the phenotype of *sgd1* (Fig. 4g–i). Moreover, the *Siubc32/sgd1* double mutant exhibited a significant inhibition in plant growth and development, with a reduction of about 66.4% in panicle length and a 43.2% decrease in 1000-grain weight compared to the WT plants (Fig. 4g–k). SEM analysis showed that the reduction in grain size in *sgd1*, *Siubc32*, and *Siubc32/sgd1* were due to a decrease in cell size but not cell number (Fig. 4j, k and Supplementary Fig. 8c). To investigate whether SGD1 and SiUBC32 function in a common pathway, we compared the agronomic traits of WT, *sgd1*, *Siubc32*, and *Siubc32/sgd1* (Fig. 4k and Supplementary Table 4). From the data, it is evident that the double mutant exhibited more severe reduction in all investigated traits compared to each single mutant. Moreover, we compared the reductions observed in the two single mutants with that in the double

mutant. The results showed that the reductions of plant height, flag leaf length, and panicle length in the double mutant were much greater than in each single mutant, suggesting that these two genes act additively in regulating these three traits. However, for grain yield-related traits, including 1000-grain weight, grain length, and grain width, the reductions observed in the double mutant were only slightly greater than those in each single mutant (Supplementary Table 4). This result suggests that SGD1 and SiUBC32 have partial overlapping functions in regulating grain yield. At the biochemical level, we also demonstrated that SGD1 and SiUBC32 interacted with each other in vivo and in vitro. However, we should recognize that SGD1 and SiUBC32 are not simply acting in the same linear pathway or as a single complex. This is supported by the fact that, in the pulldown experiment, the band of SGD1 was much weaker compared to that of SiUBC32. A previous report also suggested that UBC32 interacts with and ubiquitinates another RING-type E3 ligase, Rma1[26]. Combining our results with previous reports[20,26], we suggest that SGD1 acts, at least in part, in a common genetic pathway with SiUBC32 to control grain yield-related traits.

### *SGD1* is involved in BR signaling

In *Setaria*, two distinctive BR-related biological processes were reported, including bristle development[27] and leaf droopiness[19]. Notably, the *sgd1* mutant exhibited short bristle length (Fig. 5a, b) and compact leaf architecture (Supplementary Fig. 9), which resembled the BR-defective phenotypes reported in previous studies[19,27]. We, therefore, considered that *SGD1* may be involved in the BR-related pathway. To test this hypothesis, we investigated the response of *sgd1* to treatment with 24-epi brassinolide (eBL, a naturally-occurring active BR). First, we measured the leaf angle and leaf droopiness in *sgd1* in response to exogenous eBL. The leaf angle of WT seedlings increased from 32.1° to 46.5° (on average) upon treatment with 5 μM exogenous eBL for 3 days, while the leaf angle in *sgd1* was 34.65 ± 1.77° under the same treatment (Fig. 5c, d). In addition, exogenous eBL led to a bent leaf blade (assessed by measuring leaf droopiness index, as illustrated in Supplementary Fig. 9a) in WT but not mutant (Fig. 5c–e). Given that excessive BR represses root growth[28,29], we grew the WT and *sgd1* mutant on 1/2 MS medium containing a high concentration of eBL to measure root growth in response to BR. Treatment with 0.01 and 0.1 μM eBL significantly inhibited root growth in WT seedlings but not *sgd1* mutants (Fig. 5f, g). These results indicated that *sgd1* was less sensitive to exogenous BL. Moreover, we analyzed the expression of BR-associated genes using RNA-seq and qPCR. According to previous studies[19,30], we identified 2793 putative BZR1-target genes in foxtail millet (Supplementary Data 5). Of these genes, 53.61% were differentially expressed (absolute $\log_2$ fold-change (FC) ≥1, P ≤ 0.05) in WT plants treated with exogenous eBL compared to controls (WT plants treated with mock); 71.90% were differentially expressed in the BR hypersensitive mutant *dpy1*, and only 18.25% were responsive in *sgd1* mutants (Fig. 5h, i and Supplementary Data 5). We further compared expression values of 555 co-expressed genes identified from the

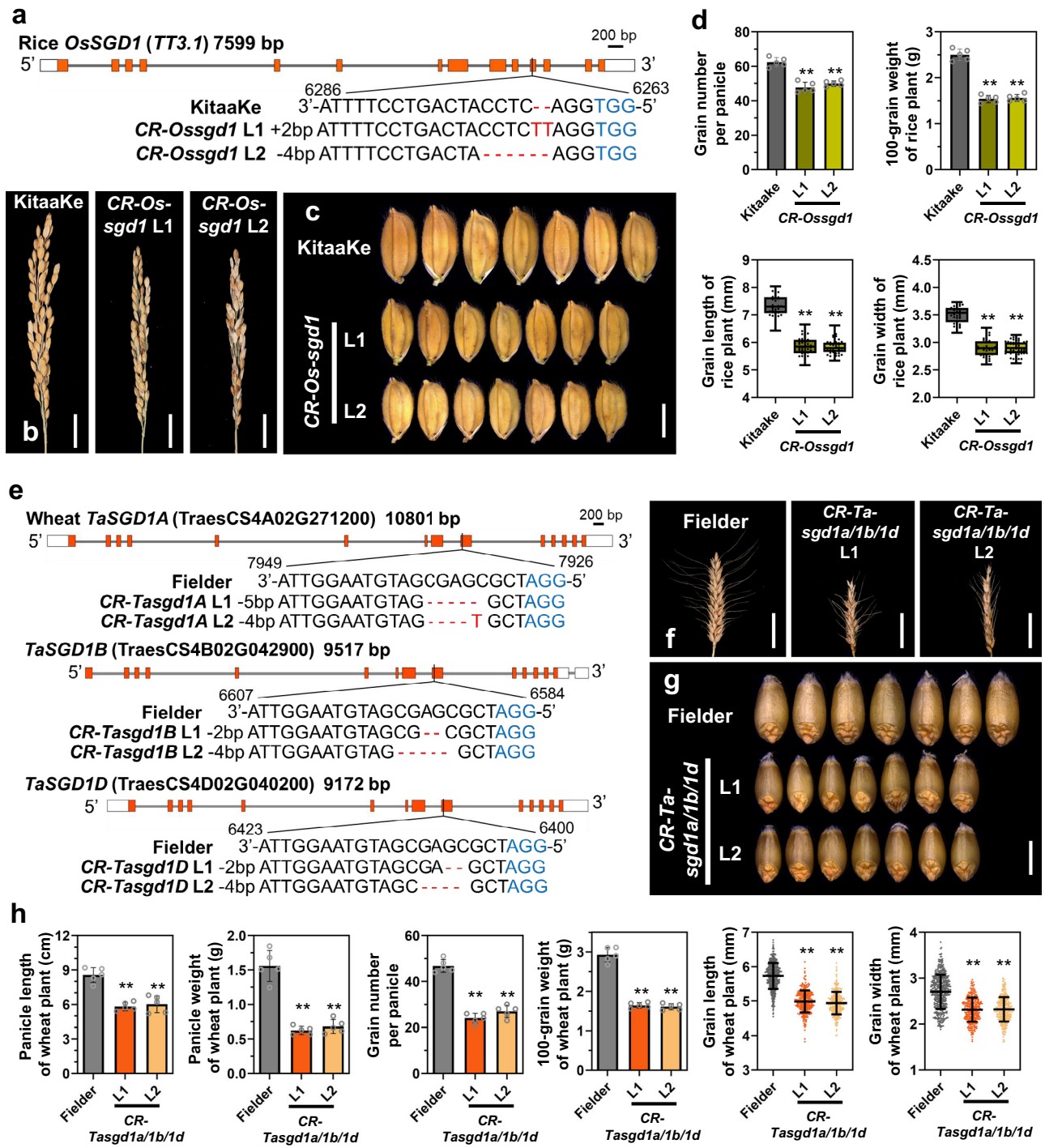

**Fig. 3 | *SGD1* regulates grain yield in major cereal crops. a** Generation of two independent rice *OsSGD1* CRISPR-edited plant lines. The *OsSGD1* gene structure, sgRNA sequence, and resulting mutations are highlighted. **b** Panicles of wild-type (WT) KitaaKe, *CR-Ossgd1*-L1, and *CR-Ossgd1*-L2 rice plants. Bar = 2 cm. **c** Mature grains of WT, *CR-Ossgd1*-L1, and *CR-Ossgd1*-L2 rice plants. Bar = 4 mm. **d** Grain number per panicle (*n* = 5), 100-grain weight (*n* = 5), and grain length and width (*n* > 15) in WT, *CR-ossgd1*-L1, and *CR-ossgd1*-L2 plants. The two ends of the box plot and the upper, middle, and lower box lines represent the upper edge, lower edge, median, and two quartiles of values in each group. Error bars indicate mean ± SD. Significant differences were determined using unpaired two-sided Student's *t*-tests.

\*\**P* < 0.001 vs. WT plants. **e** Generation of two independent wheat *TaSGD1A/1B/1D* triple mutants by CRISPR/Cas9. The gene structures, sgRNA sequences, and resulting mutations of *TaSGD1A*, *TaSGD1B*, and *TaSGD1D* lines are illustrated. **f** Panicles of WT (Fielder), *CR-Tasgd1a/1b/1d*-L1, and *CR-Tasgd1a/1b/1d*-L2 plants. Bar = 3 cm. **g** Mature grains of WT (Fielder), *CR-Tasgd1a/1b/1d*-L1, and *CR-Tasgd1a/1b/1d*-L2 lines. Bar = 4 mm. **h** Panicle length and weight (*n* = 5), grain number per panicle (*n* = 5), 100-grain weight (*n* = 5), and grain length and width (*n* > 100) in the plant lines described in (**e**). Error bars indicate mean ± SD. Significant differences were determined using unpaired two-sided Student's *t*-tests. \*\**P* < 0.001 vs. WT plants. Source data are provided as a Source Data file.

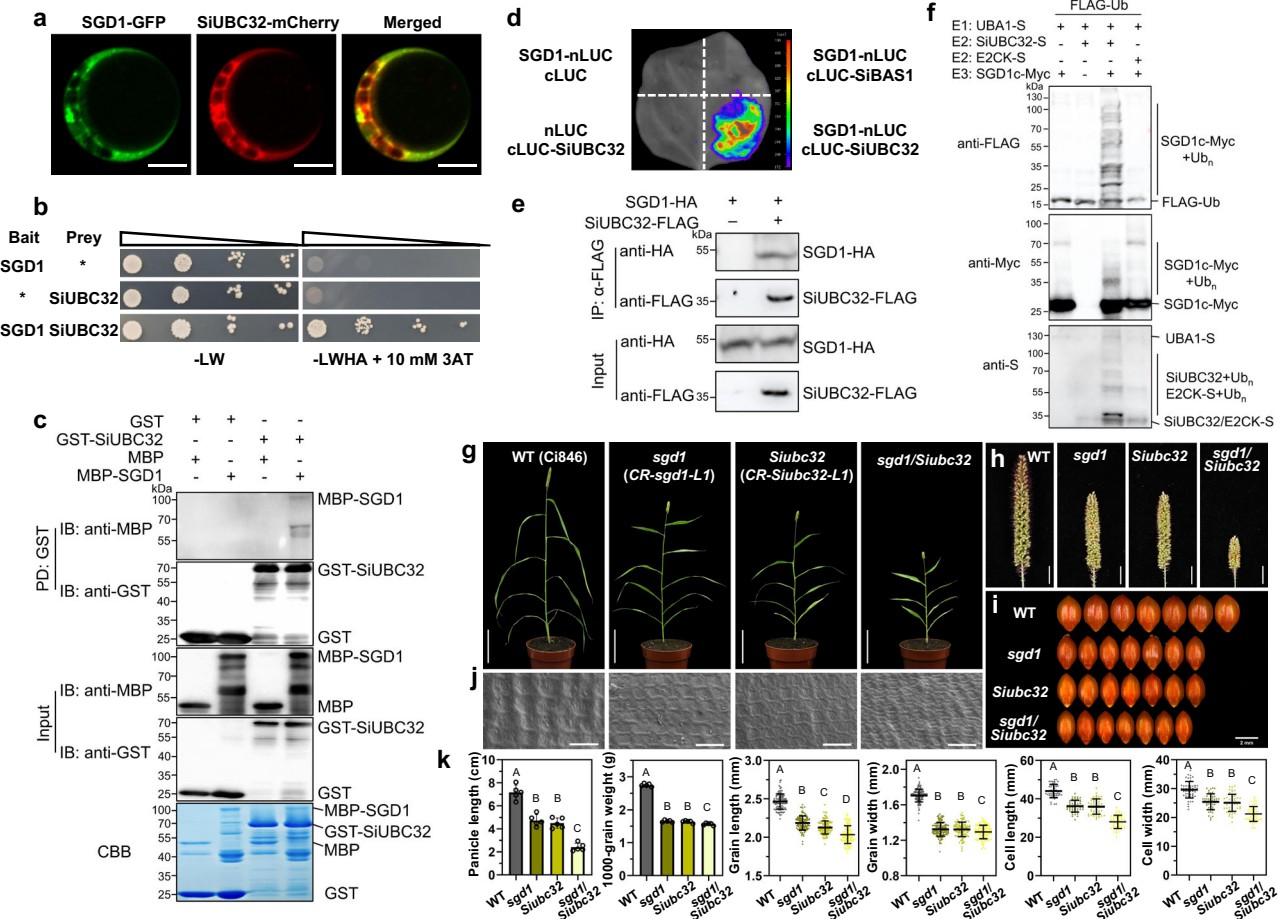

**Fig. 4 | SiUBC32 directly interacts with SGD1 and regulates grain yield.**
**a** Subcellular localization of SGD1 and SiUBC32 in *S. italica* protoplasts. Bar = 10 μm.
**b** Split-ubiquitin membrane-based yeast two-hybrid analysis of the interaction between SGD1 and SiUBC32. The asterisks represent empty vector controls. Positive interactions were evaluated using yeast cells grown on a synthetic defined medium lacking Leu, Trp, His, and adenine (−LWHA) in the presence of 10 mM 3-aminotriazole (3-AT). **c** In vitro pull-down analysis of the interaction between MBP-SGD1 and GST-SiUBC32. GST-tagged proteins were immobilized on glutathione sepharose beads and incubated with maltose-binding protein (MBP)-tagged proteins. Washed beads were immunoblotted with anti-MBP or anti-GST (top two panels). Input proteins are shown by immunoblotting (middle two panels) and Coomassie blue (CBB) staining (bottom). **d** SGD1 interacts with SiUBC32 in the split luciferase complementation assay. SiBAS1 was used as a negative control. Vectors were paired and co-transformed into tobacco leaves. **e** Analysis of the SGD1–SiUBC32 interaction using in vivo co-immunoprecipitation (Co-IP) assay. SGD1-HA and SiUBC32-FLAG were co-expressed in *S. italica* protoplasts. IP was performed using anti-FLAG antibodies, and the associated protein was detected by immunoblotting with anti-HA antibodies. **f** The E3 ligase activity of SGD1. UBA1-S, SiUBC32-S, E2CK-S, SGD1c-Myc, and His-FLAG-Ub were expressed in *E. coli*. UBA1, SiUBC32, and E2CK were detected by immunoblotting using anti-S antibodies. SGD1c activity was detected by anti-Myc antibodies. Ub conjugates were detected by anti-FLAG antibodies. These experiments from (**a**–**j**) were repeated three times independently with similar results. **g** Morphological features of wild type (WT), CRISPR-edited *sgd1* (*CR-sgd1*-L1), *Siubc32* (*CR-Siubc32*-L1), and *sgd1*/*Siubc32* grown in a growth chamber for 40 days under a 10-h light/14-h dark cycle. Bar = 10 cm. **h** Panicles of WT, *sgd1*, *Siubc32*, and *sgd1*/*Siubc32* plants. Bar = 1 cm. **i** Grain size of WT, *sgd1*, *Siubc32*, and *sgd1*/*Siubc32* lines. Bar = 2 mm. **j** SEM analysis of the lemmas of WT, *sgd1*, *Siubc32*, and *sgd1*/*Siubc32* plants. Bar = 60 μm. This experiment was repeated five times independently with similar results. **k** Panicle length ($n = 5$), 1000-grain weight, grain length and width ($n > 100$), and cell length and width ($n = 50$) of WT, *sgd1*, *Siubc32*, and *sgd1*/*Siubc32* plants. Error bars indicate mean ± SD. Different lowercase letters indicate significant differences ($P < 0.05$, one-way analysis of variance with Tukey's multiple comparisons test). Source data are provided as a Source Data file.

WT-BR, *dpy1*, and *sgd1* samples (Supplementary Data 6); the result suggested 81.2% of these significant differentially expressed genes (DEGs) in WT-BL and *dpy1* turned into no-DEGs (|log2FC| <1) in *sgd1*, and 11.2% of the genes showed an inversed expression pattern. The transcriptional feedback regulation of BR biosynthetic genes is mediated by BR signaling[29]. Thus, we measured the relative expression levels of *SiD2*[28] and *SiCYP51G3*[31], which play an essential role in BR biosynthesis. qPCR results showed that the expression of *SiD2* and *SiCYP51G3* was significantly higher in *sgd1* mutants than in WT plants (Fig. 5j). In addition, the transcription of two BR-induced genes (*GLR2.7* and *CBF2*) and one BR-repressed gene (*BRH1*)[32–34] were analyzed in WT and *sgd1* plants under eBL treatment. RNA-seq (Supplementary Data 6) and qPCR (Fig. 5k) revealed that treatment with 0.01 μM eBL upregulated BR-induced genes and downregulated BR-repressed genes more

strongly in WT plants than in *sgd1* mutants in response to BR. Taken together, our results suggested that the *sgd1* mutant had decreased sensitivity to BR, and therefore supported the involvement of the *SGD1* gene in BR signaling.

## SGD1 interacts with BRI1 and contributes to its stabilization

We assessed the interactions of SGD1 with candidate proteins that play crucial roles in BR signaling, including BRI1, BAK1, and BIN2, using LCA and membrane-based Y2H assay. Positive luciferase signals were detected in the SGD1-BRI1 pair (Fig. 6a). We then tested whether SGD1 interacted with BRI1 using Y2H assays. Since BRI1 often works with its partners through C-terminal kinase domain[19], we separated SiBRI1 into an N-terminal transmembrane domain (SiBRI1n) and a C-terminal kinase domain (SiBRI1c) to investigate their interactions with SGD1

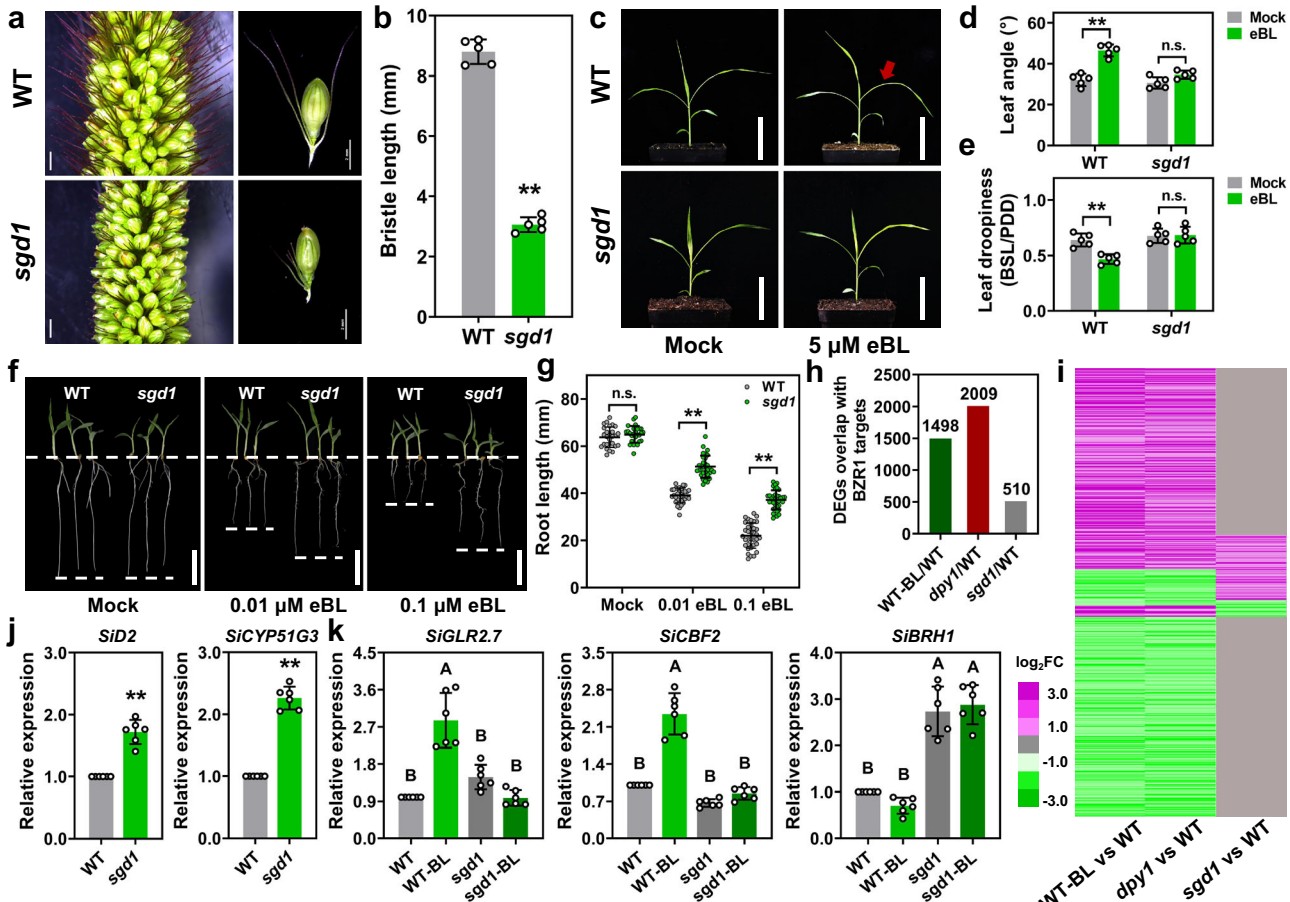

**Fig. 5 | The *sgd1* mutant shows decreased sensitivity to BR. a** Bristle phenotype in wild-type (WT) and *sgd1* plants. Bar = 2 mm. **b** Comparison of bristle length between WT and *sgd1* lines ($n = 5$, unpaired two-sided Student's *t*-tests, **$P < 0.001$. Error bars indicate mean ± SD. **c** Leaf architecture responses to BR in WT and *sgd1* plants. Plants were treated with 5 µM 24-epi-brassinolide (eBL) for 3 days. **d, e** Leaf angle (**d**) and leaf drooping (**e**) of the second leaf from the top as red arrows indicated in (**c**). Leaf drooping was calculated by the bending site length (BSL) to proximal-distal distance (PDD) ratio ($n = 5$, unpaired two-sided Student's *t*-tests, **$P < 0.001$; n.s. means not statistically significant; Error bars indicate mean ± SD). **f** Root growth phenotypes in WT and *sgd1* plants grown in the presence of the indicated concentration of eBL for 6 days in a growth chamber under a 10-h light/14-h dark cycle. Bar = 2 cm. **g** Root length in WT and *sgd1* plants under eBL treatment ($n > 25$, unpaired two-sided Student's *t*-tests, **$P < 0.001$; n.s. means not statistically significant; Error bars indicate mean ± SD). **h** Number of differentially expressed genes overlap with BZR1 targets in WT-BL/WT, *dpy1*/WT, and *sgd1*/WT plants. **i** Heatmap of the expression patterns of BZR1-target genes (Supplementary Data 6) in WT-BL/WT, *dpy1*/WT, and *sgd1*/WT lines. **j** qRT-PCR analysis of the relative expression levels of *SiD2* and *SiCYP51G3* in 7-day-old WT and *sgd1* seedlings, $n = 6$, two-sided Student's *t*-tests, **$P < 0.001$. Error bars indicate mean ± SD. **k** Relative expression levels of *SiGLR2.7*, *SiCBF2*, and *SiBRH1* in WT and *sgd1* plants treated with 0.01 µm eBL. Gene expressions were quantified by qRT-PCR with six replications. Error bars indicate mean ± SD. Different lowercase letters indicate significant differences ($P < 0.05$, one-way analysis of variance with Tukey's multiple comparisons test). Source data are provided as a Source Data file.

using Y2H. The result showed that SGD1 interacted with full-length SiBRI1 and SiBRI1c but not with SiBRI1n in yeast (Fig. 6b). Then, we assessed the interactions between SGD1 and BRI1 in a pull-down assay using maltose-binding protein (MBP)-SGD1 and GST-SiBRI1c fusion proteins purified from *E. coli*. The GST pull-down assay showed that only GST-SiBRI1c bound to MBP-SGD1 (Fig. 6c), suggesting a direct interaction between SGD1 and SiBRI1 in vitro. We further assessed this interaction using an in vivo Co-IP assay in foxtail millet protoplasts. The results showed that SiBRI1-HA was present in SGD1-FLAG immunoprecipitates (Fig. 6d). Collectively, we demonstrate that SGD1 interacts with SiBRI1 in vivo and in vitro.

E3 ligases bind to substrates and target them for ubiquitination. Therefore, we investigated whether SGD1 can ubiquitinate SiBRI1 using an in vitro ubiquitination assay. Ubiquitination was observed in recombinant MBP-SiBRI1c-HA protein in the presence of ubiquitin, E1, E2, and SGD1c, but not with the RING domain disturbed mSGD1c, indicating that SGD1 can ubiquitinate SiBRI1 in vitro (Fig. 6e). We further investigated the ubiquitination of SiBRI1 in WT and *sgd1* mutant by in vivo ubiquitination assays.

SiBRI1-HA and FLAG-Ub (ubiquitin tagged with the FLAG epitope) were co-expressed in protoplasts of WT and mutant plants, respectively. After immunoprecipitation with anti-FLAG antibodies, SiBRI1 protein ubiquitination was analyzed by immunoblotting with anti-HA antibodies. Notably, we detected ubiquitination of SiBRI1 proteins in both WT and *sgd1* plants, but the ubiquitination level of SiBRI1 was much lower in the mutant compared to that of the WT (Fig. 6f). To determine whether SiBRI1 ubiquitination by SGD1 led to protein degradation, we quantified SiBRI1 protein levels by transiently expressing both the catalytically inactive (mSGD1) and active form of SGD1 with BRI1 in foxtail millet protoplasts. Interestingly, the catalytically active form of SGD1 increased BRI1 protein levels, while mSGD1 did not affect BRI1 accumulation (Fig. 6g). The result suggests that SGD1 ubiquitinates SiBRI1 for protein stabilization rather than degradation, and this type of stabilization is dependent on the catalytic activity of SGD1. We also verified this result in transgenic plants. Total protein was extracted from Ci846, *sgd1*, and SGD1-GFP overexpression plants (*OE-SGD1*). SiBRI1 proteins was detected

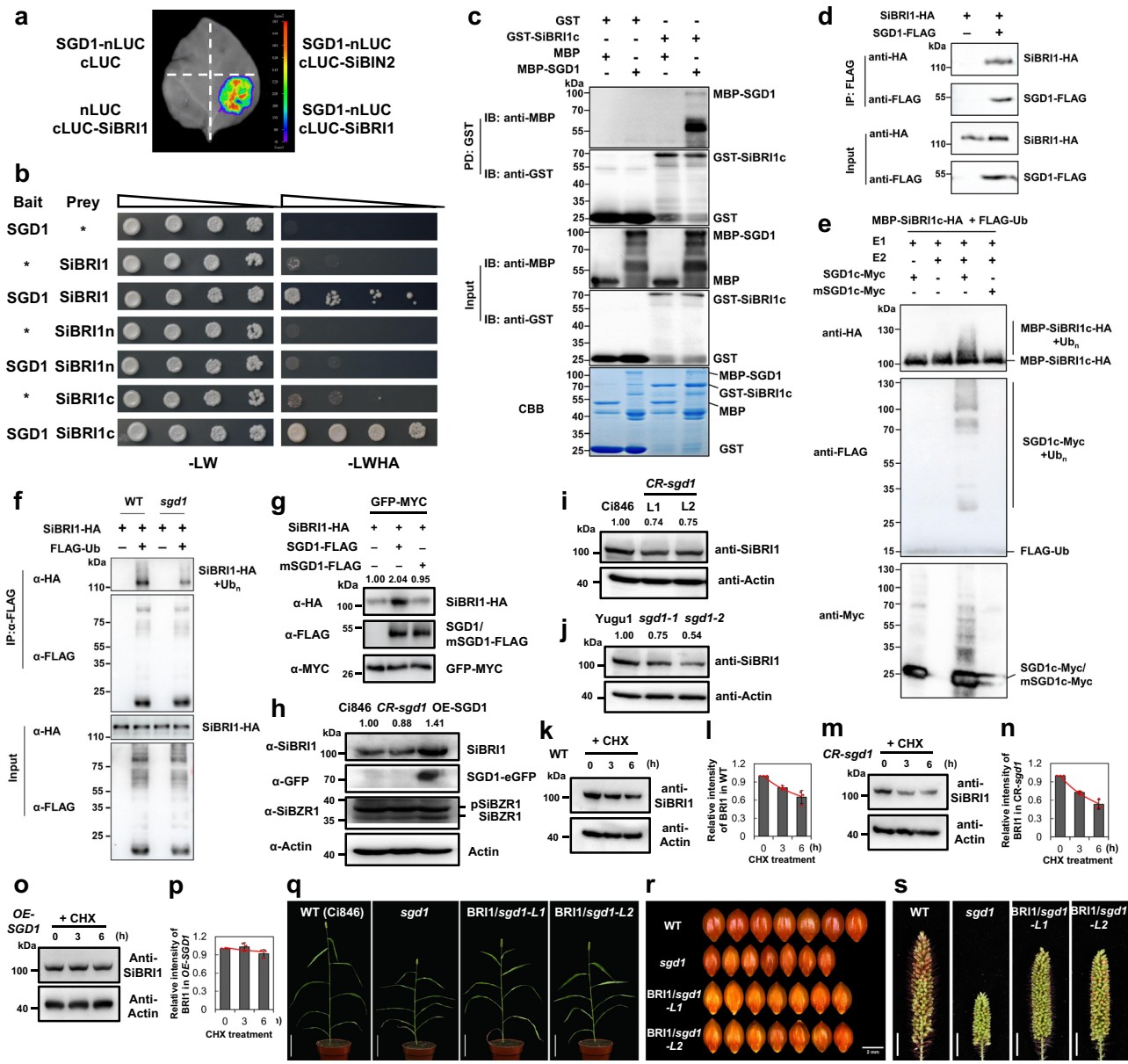

using foxtail millet anti-SiBRI1 antibodies prepared in our previous study[19]. The specificity of the SiBRI1 antibody was verified and demonstrated in Supplementary Fig. 10. BRI1 accumulated in *SGD1*-overexpressing plants, demonstrating that SGD1 did not induce BRI1 degradation (Fig. 6h). BRASSINAZOLE-RESISTANT1 (BZR1) is the key transcription factor in the BR signaling pathway. The phosphorylated and dephosphorylated levels of BZR1 can serve as indicators of BR signaling output. We found that the levels of BRI1 and dephosphorylated SiBZR1 increased in *OE-SGD1* plants and decreased in *sgd1* mutants. Meanwhile, the levels of phosphorylated SiBZR1 decreased in *OE-SGD1* plants and increased in *sgd1* mutants. These results suggest that BR signaling was enhanced in *SGD1*-overexpressing plants and repressed in *sgd1* mutants (Fig. 6h). This result is also consistent with the BR-hyposensitive phenotypes of *sgd1* mutants. We also compared SiBRI1 abundance in four *sgd1* mutant lines (*CR-sgd1*-L1/L2 under a Ci846 background, *sgd1-1/2* under a Yugu1 background), and found that BRI1 abundance was lower in all examined *sgd1* mutants than in WT plants, indicating that the loss of function of SGD1 led to reduced BRI1 stability (Fig. 6i, j and Supplementary Fig. 11). Moreover, we performed cycloheximide (CHX)

treatments. SiBRI1 protein stability in WT (Ci846), *CR-sgd1* mutants, and *OE-SGD1* plants was assessed by monitoring protein abundance over a 6-h period after CHX treatment. The relative intensities of the SiBRI1 protein level were measured based on three independent biological replicates. CHX time-dependently decreased SiBRI1 levels in WT and mutant plants, and the decrease was more pronounced in the mutant lines (Fig. 6k–n and Supplementary Fig. 12a, b). It is worth noting that in the presence of CHX, the level of SiBRI1 remained stable in SGD1-overexpressing lines, with only a slight decreasing trend observed (Fig. 6o, p and Supplementary Fig. 12c). To sum up, these results indicate that SGD1 ubiquitinates and stabilizes SiBRI1.

To determine the relationship between SGD1 and SiBRI1, we conducted genetic complementary experiments by overexpressing *SiBRI1-GFP* to the *sgd1* mutant background. Two independent positive transgenic lines (SiBRI1/*sgd1*-L1 and SiBRI1/*sgd1*-L2) were obtained (Supplementary Fig. 13a, b). The comparison of major agronomic traits among BRI1/*sgd1*, *sgd1*, and WT plants showed that SiBRI1 overexpression in *sgd1* can partially rescue the mutant phenotype (Fig. 6q–s and Supplementary Fig. 13c), which positively supported that SGD1 promotes BR signaling through enhancing the stability of SiBRI1.

**Fig. 6 | SGD1 interacts with SiBRI1 and enhances SiBRI1 protein stability. a** Split luciferase complementation assay of the interaction between SGD1 and SiBRI1. The indicated vector pairs were co-transformed into tobacco leaves. SiBIN2 was used as a negative control. **b** Split-ubiquitin membrane-based yeast two-hybrid analysis of the interaction between SGD1 and SiBRI1. SiBRI1, SiBRI1n, and SiBRI1c represent the full-length, N-terminal transmembrane, and C-terminal kinase domain, respectively. Asterisks indicate empty vectors. Positive interactions were evaluated using yeast cells grown on a synthetic defined medium lacking Leu, Trp, His, and adenine (−LWHA). **c** In vitro pull-down analysis of the interaction between SGD1 and SiBRI1c. GST or GST-SiBRI1c were immobilized on glutathione sepharose beads and incubated with maltose-binding protein (MBP) or MBP-SGD1. Washed beads were immunoblotted with anti-MBP or anti-GST antibodies (upper two panels). Input immunoblotted proteins are shown in the third and fourth panels, and CBB-stained proteins are shown in the bottom panel. **d** In vivo Co-IP analysis of the interaction between SGD1 and SiBRI1. SiBRI1-HA and SiBRI1-FLAG were co-expressed in *S. italica* protoplasts. Co-IP was performed using anti-FLAG antibodies, and the associated protein was detected by immunoblotting with anti-HA antibodies. **e** Ubiquitination of SiBRI1 by SGD1. MBP-SiBRI1-HA, E1, E2, SGD1c-Myc, mSGD1c-Myc (C426A and H443A), and His-FLAG-Ub were expressed in *E. coli*. SiBRI1 ubiquitination was detected by immunoblotting with anti-HA antibodies. Ub conjugates were detected using anti-FLAG antibodies. SGD1c and mSGD1c activity was detected with anti-Myc antibodies. **f** SiBRI1 ubiquitination level was lower in *sgd1* than in WT plants. SiBRI1-HA and FLAG-Ub were co-expressed in WT and *sgd1* protoplasts. Following IP with anti-FLAG, SiBRI1 ubiquitination was detected by immunoblotting with anti-HA antibodies. **g** SGD1 enhances SiBRI1 protein stability in *S. italica* protoplasts. SiBRI1-HA, SGD1-FLAG, and mSGD1-FLAG were detected with anti-HA

and FLAG antibodies. GFP-Myc was used as an internal transfection control and detected using anti-Myc antibodies. The relative abundance of SiBRI1-HA is shown above the blot. **h** SGD1 enhances SiBRI1 protein stability in vivo. Fourteen-day-old seedlings collected from Ci846, *sgd1*, and *SGD1*-overexpressing plants (*OE-SGD1*) were used for immunoblots. SiBRI1, SGD1-eGFP, and SiBZR1 (phosphorylated and dephosphorylated) were detected by immunoblotting with antibodies against SiBRI1, GFP, and SiBZR1. Actin was used as a loading control. **i** SiBRI1 stability was reduced in CRISPR-edited *sgd1* mutant lines. SiBRI1 in 14-day-old Ci846 (WT), *CR-sgd1*-L1, and *CR-sgd1*-L2 seedling leaves was detected by immunoblotting with anti-SiBRI1 antibodies. **j** SiBRI1 stability was reduced in EMS-induced *sgd1* mutant lines. SiBRI1 in Yugu1 (WT), *sgd1-1*, and *sgd1-2* seedling leaves were detected by immunoblotting with an anti-SiBRI1 antibody. The relative abundance of SiBRI1 in (**h**–**j**) is shown above the blot. These experiments in (**a**–**h**) were repeated three times independently with similar results. **k**–**p** SiBRI1 concentration in 14-day-old Ci846 (WT, **k**), *sgd1* (*CR-sgd1*, **m**), and *OE-SGD1* (**n**) seedling leaves treated with 100 μM cycloheximide (CHX) for 0, 3, and 6 h. SiBRI1 was detected using anti-SiBRI1 antibodies. Actin was used as a loading control. Three biological replicates were used for SiBRI1 protein abundance measurements (Supplementary Fig. 12). Quantitation of the relative SiBRI1 immunoblot signals was shown in (**l**, **n**, **p**), $n = 3$ biological replications. Error bars indicate mean ± SD. **q**–**s** Overexpression of *SiBRI1* partially rescues the *sgd1* phenotype. **q** Morphological features of WT (Ci846), *sgd1*, BRI1/*sgd1*-L1, and BRI1/*sgd1*-L2 (transformation of *pUbi:SiBRI1-eGFP* to an *sgd1* background) plants are grown in a growth chamber for 40 days under a 10-h light/14-h dark cycle. Bar = 10 cm. **r** Grain size of WT, *sgd1*, BRI1/*sgd1*-L1, and BRI1/*sgd1*-L2 lines. Bar = 2 mm. **s** Panicles of WT, *sgd1*, BRI1/*sgd1*-L1, and BRI1/*sgd1*-L2 plants. Bar = 1.5 cm. Source data are provided as a Source Data file.

## Overexpressing the elite haplotype of *SGD1* improves the grain yield of foxtail millet

To detect selective signatures of *SGD1* in *Setaria* domestication and improvement, we collected high-throughput resequencing data of diversified collections of 1681 *Setaria* germplasm, including 457 *S. italica* cultivars, 694 landraces[35,36], and 530 *S. viridis* (wild ancestor of foxtail millet) accessions[37]. Whole-genome selective sweep analysis was performed using the fixation index (*Fst*) and nucleotide diversity value (π), respectively. A previously reported domesticated gene *qSh1* (controls seed shattering)[36] was found in the candidate region, indicating our analysis was reliable. To our interest, the genomic region containing the *SGD1* locus ranked among the top 1% of the empirical *Fst* (*Fst* ≥0.607) distribution in the comparison between wild ancestors and landraces (Fig. 7a, b), indicating that *SGD1* was under selection during the domestication process. Moreover, *SGD1* was present in selective sweeps that passed the thresholds of π ratio between landraces and cultivars (the 1% right tail of the empirical π ratio distribution, π ratio ≥4.217), suggesting that *SGD1* also underwent selection in foxtail millet breeding improvement (Fig. 7c). A more detailed investigation of π distributions in the gene body of *SGD1* revealed that selections happened mainly in the promoter and intron regions, while nucleotide sequences in exons were highly conserved, especially in the RING finger domain (Fig. 7d).

Thirteen agronomic traits related to grain yield, including grain weight per main panicle, 1000-grain weight, grain length, and grain width, were investigated in the *Setaria* germplasm collections (Supplementary Table 5). Haplotype analysis suggested that *SGD1* had four main haplotypes in these collections (Supplementary Fig. 14a, b). The analysis of the distribution of the four haplotypes in different *Setaria* germplasm pools revealed strong domestication and improvement of *SGD1* (Fig. 7e and Supplementary Fig. 14c). In wild species, over 98% of the varieties carried the H2 haplotype, which was associated with the lowest yield performance. During domestication, the frequency of H2 haplotypes decreased to 3.7%, while the frequencies of the H1, H3, and H4 haplotypes increased, becoming predominant in landraces with frequencies of 61.0, 23.1, and 12.2%, respectively. Notably, the H1 haplotype was found to be the most beneficial for crop production and was consequently selected, leading to its frequency increasing to

70.5% in modern cultivars (Fig. 7e). Association analysis of major agronomic traits in different haplotypes demonstrated that H1 of *SGD1* was associated with higher panicle weight and grain weight and a higher number of seeds per panicle (Fig. 7f), supporting the fact that H1 of *SGD1* (*SGD1^H1*) is the high grain yield haplotype selected by farmers and breeders. Interestingly, we also found that *SGD1* was associated with foxtail millet blast disease resistance, suggesting that *SGD1* may be involved in crop disease resistance (Supplementary Fig. 14d). We thus overexpressed *SGD1^H1* driven by the *ubiquitin-1* promoter (*Ubi::SGD1^H1*) into foxtail millet Ci846 (an H3 background) and obtained two positive independent overexpression lines (Supplementary Fig. 15a). The comparison of grain yield-related traits in a field experiment showed that *SGD1^H1*-overexpressing lines had a higher grain area, 1000-grain weight, and panicle length than WT lines, resulting in a 12.82% increase in grain yield per plant (Fig. 7g–k and Supplementary Fig. 15b). In addition, we found that overexpression lines exhibited resistance to blast disease in the field after inoculated with the *Pyricularia setariae* race (Supplementary Fig. 15c), which supports the result of the haplotype analysis.

RNA-sequencing analysis of WT, *sgd1*, and *OE-SGD1^H1* plants provided additional information on the role of *SGD1* in promoting grain yield. A total of 2599 DEGs were identified between *sgd1* and Ci846 and 436 DEGs between *OE-SGD1^H1* and Ci846 (Supplementary Fig. 15d). In addition to BR signaling pathway that positively mediated by *SGD1*, we noted that these DEGs were significantly enriched in biological processes including protein processing in ER, photosystem II (PS II) stabilization, response to heat and other stresses, defense response to a bacterium, chlorophyll biosynthesis, nitrogen metabolism, and organ growth regulation (Supplementary Fig. 15e, f). Moreover, it has been demonstrated that rice *SGD1* (*TT3.1*) increases rice grain yield under heat stress by protecting chloroplasts and maintaining photosynthesis[20]. Combining our findings with previous reports, we conclude that the increase in grain yield may be a result of *SGD1*'s comprehensive effect on plant growth and stress responses. Further investigations are required to better understand the underlying mechanism of *SGD1* in integrating the regulation of multiple pathways.

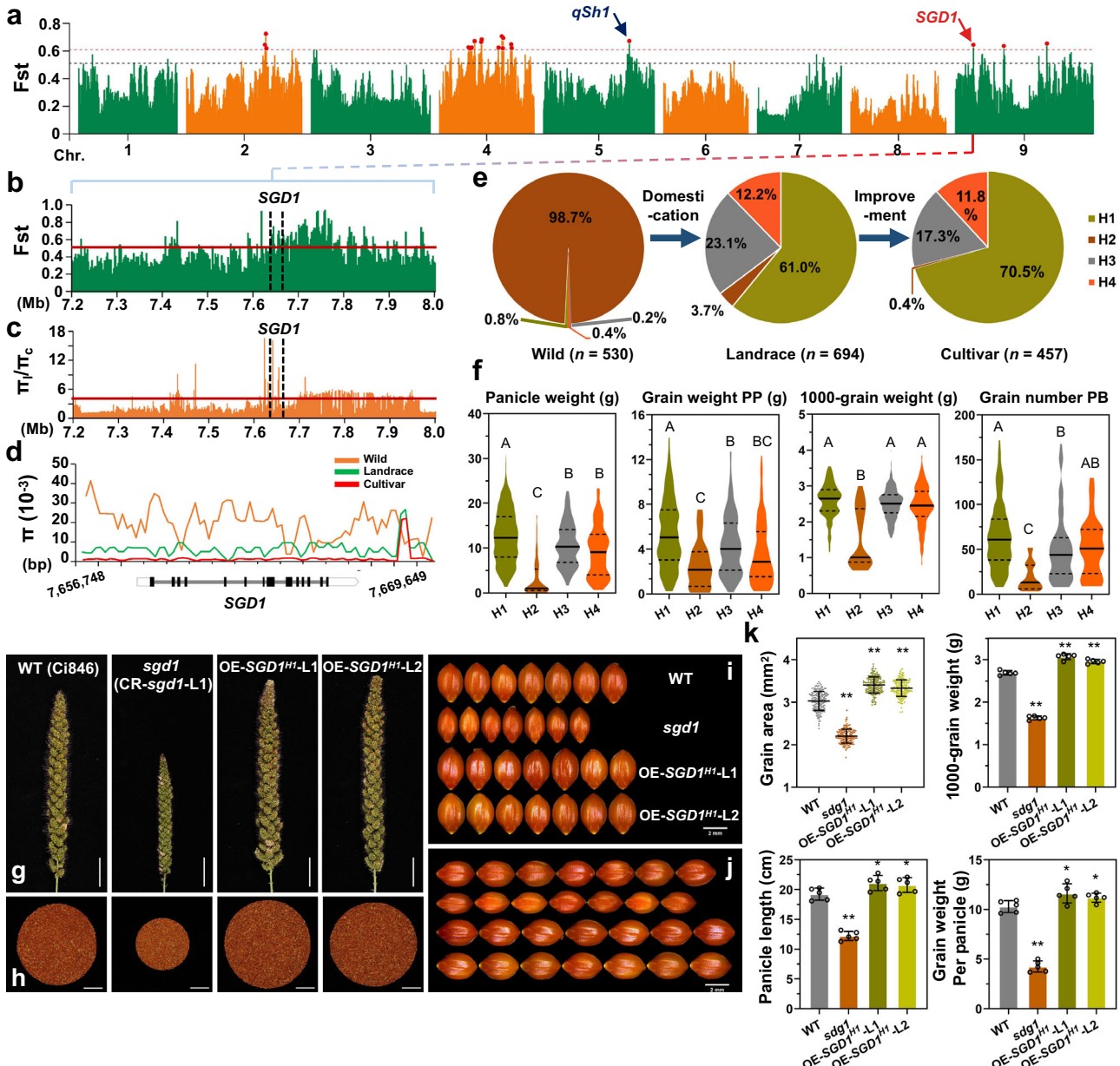

**Fig. 7 | Natural variations of *SGD1* associated with grain yield improvement in *Setaria*. a** Whole-genome selective sweep analysis using the *Fst* index. The black and red dashed lines indicate 5 and 1% thresholds, respectively. The red dots indicate potential genomic regions that met the 1% threshold. **b** *Fst* of *SGD1* and a neighboring 400-kb region between wild species and landraces. The red line indicates a 5% threshold. **c** The nucleotide diversity index (π) of *SGD1* and a neighboring 400-kb region between landraces and cultivars. The red line indicates a 5% threshold. The vertical dashed lines in (**b**, **c**) represent the *SGD1* genomic region. **d** π in the *SGD1* gene in different *Setaria* subgroups. The position and gene structure of *SGD1* are illustrated in the Y axis. **e** Haplotype analysis revealed that *SGD1* was under human selection. Four *SGD1* haplotypes are shown in different colors. *n* corresponds to the total number of varieties in each pool. **f** Major agronomic traits (panicle weight, grain weight per panicle, 1000-grain weight, and grain number per branch) were analyzed in 960 of 1681 *Setaria* germplasms by *SGD1*

haplotype. Uppercase letters indicate significant differences (*P* < 0.05, one-way analysis of variance with Tukey's multiple comparisons test). **g–j** Morphological features of wild type (WT), *sgd1*, *OE-SGD1^{H1}*-L1, and *OE-SGD1^{H1}*-L2 (two independent transgenic lines overexpressing *SGD1^{H1}* in WT background). **g** Panicles of WT, *sgd1*, *OE-SGD1^{H1}*-L1, and *OE-SGD1^{H1}*-L2. Bar = 3 cm. **h** Grains per panicle in WT, *sgd1*, *OE-SGD1^{H1}*-L1, and *OE-SGD1^{H1}*-L2 (from left to right) plants. Bar = 3 cm. **i** Grain width in WT, *sgd1*, *OE-SGD1^{H1}*-L1, and *OE-SGD1^{H1}*-L2 plants. Bar = 2 mm. **j** Grain length in WT, *sgd1*, *OE-SGD1^{H1}*-L1, and *OE-SGD1^{H1}*-L2 (from top to bottom) lines. Bar = 2 mm. **k** Measurements of grain area, 1000-grain weight, panicle length, and grain weight per panicle in WT, *sgd1*, *OE-SGD1^{H1}*-L1, and *OE-SGD1^{H1}*-L2 plants. *n* = 170 for grain area measurement and *n* = 5 for other measurements. Data were means ± SD. Significant differences were determined using unpaired two-sided Student's *t*-tests. *\*P* < 0.05, \*\**P* < 0.01 vs. WT plants). Source data are provided as a Source Data file.

## Discussion

Identification of key genes involved in grain yield improvement is of vital importance for agricultural production. Protein ubiquitination is involved in grain yield regultion[2]. Specifically, four ubiquitin receptors and five E3 ubiquitin ligases are implicated in seed size control in plants[2]. However, as a core component of ubiquitination, only a few E2

ubiquitin-conjugating enzymes were reported to regulate grain yield[38]. Our study identified an important RING-type E3 ligase SGD1 that is involved in regulating grain weight, grain size, and panicle size. Additionally, we identified SiUBC32 as the E2 partner for SGD1. Further biochemical and genetic analysis confirmed that the E2 (SiUBC32)-E3 (SGD1) pair functioned in the ubiquitination cascade pathway and

played a crucial role in regulating grain yield. E2-E3 pairs regulating essential biological processes have been extensively studied in mammals[39] but not in plants. Our research adds to an emerging story that an E2-E3 pair impacts panicle and seed development, in addition to flowering time[40] and stress responses[26] in plants.

Moreover, we found that the E3 ligase SGD1 directly interacted with and ubiquitinated the key BR receptor BRI1. Interestingly, this interaction and ubiquitination process led to the accumulation of BRI1 and enhancement of BR signaling rather than degradation of BRI1 for the following reasons. First, accumulation of BRI1 was detected when SGD1 was overexpressed in foxtail millet protoplasts and in *SGD1*-overexpressing plants. Second, dephosphorylated BZR1 (a marker of BR signaling activation)[41] was repressed in *SGD1*-knockout plants and enhanced in *SGD1*-overexpressing plants. Finally, the *sgd1* mutant showed BR-hypo-sensitive phenotypes (Fig. 5) and reduced BRI1 stability (Fig. 6i, j). SiBRI1 stability is higher in plants overexpressing *SGD1* than in mutants and WT plants (Fig. 6k–p). In addition, *BRI1* overexpression can partially complement the *sgd1* mutant phenotype. We conclude that in addition to promoting protein degradation, ubiquitin modification of BRI1 by SGD1 may participate in non-degradation functions such as enhancing chaperone activity[42] or mediating protein trafficking[43].

However, it will likely be challenging to understand mechanistically how SGD1 keeps BR signaling at optimal efficiency through the fine-tuning of BRI1 levels. In our view, there would be two potential mechanisms. One is that SGD1 may be implicated in BRI1 trafficking. BRI1 is synthesized in the ER and transported to the Golgi apparatus for modification and sorting[44]. These processes involve membrane-bound organelles such as the ER, Golgi apparatus, and trans-Golgi network/early endosome (TGN/EE), and require the involvement of ubiquitination of the target proteins[45]. In our study, we observed a significant increase in BRI1-GFP intracellular fluorescence signals compared to that of the PM in the root of BRI1-GFP/*sgd1* plants (Supplementary Fig. 16). Moreover, several SNAREs (soluble N-ethylmaleimide sensitive factor attachment protein receptors), including SYP61 and VAMP726, were identified as putative SGD1-interacting proteins (Supplementary Data 4). Combining our findings with the report that SNAREs mediate BRI1 trafficking to the PM[46], we speculate that SGD1 may work with SNAREs to ensure proper trafficking of BRI1s from the ER to the PM, where they can execute their BR-receptor function. In addition, PM-resident BRI1 undergoes internalization through the endocytic pathway[45]. This process is also regulated by ubiquitination[18,45]. Internalized BRI1 may be recycled to the PM or the late endosomes/multivesicular bodies (MVB) for eventual degradation. A few ubiquitination sites in BRI1 associated with endocytic sorting and vacuolar targeting have been identified[45]. Two E3 ligases PUB12/13, which can mediate BRI1 polyubiquitination and promote BRI1 endocytosis and degradation, were identified[18]. In the *Arabidopsis* *pub12*/*13* mutant, BRI1 abundance and its residence time at the PM increased, while the endosomal pool of BRI1 was reduced[18]. In our study, SGD1 ubiquitinated and stabilized BRI1. The endosomal pool of BRI1 increased significantly in the *sgd1* mutant. These results suggest that SGD1 counteracted the effect of PUB12/13 on BRI1. Whether SGD1 collaborates with PUB12/13 or modifies the ubiquitination sites of BRI1 to regulate its endocytosis and degradation in foxtail millet remains unclear and will be the focus of our future study.

The other potential mechanism is that SGD1 may regulate the quality control[47] of BRI1 at the ER. A protein quality control system consists of proteases and molecular chaperones, which prevent the accumulation of misfolded proteins by refolding and degradation[47]. Transmembrane proteins like BRI1 are synthesized in the ER. A few abnormal proteins (e.g., unfolded or misfolded BRI) would also be produced, which are recognized and ubiquitinated by ER-associated degradation (ERAD)[48]. Correspondingly, as the E2 partner of SGD1, UBC32 has been shown to be an important component of ERAD in *Arabidopsis*[24]. Moreover, the RNA-seq analysis of *SGD1* loss-of-function

mutants showed that 69 DEGs involved in "protein processing in ER", and "response to unfolded protein" were enriched in mutant. Key ERAD marker genes[49] such as *Seita.1G117600* homolog to *BIP*, *Seita.7G083500* (*CNX1*), *Seita.9G023700* (*CART1a*), *Seita.2G225600* (*PDIA6*), and *Seita.3G109800* (*DER1*) were significantly upregulated in the mutant, indicating a genetic feedback loop in ERAD induced by *SGD1* mutations (Supplementary Data 7). Therefore, the SiUBC32-SGD1 complex may be responsible for the quality control of BRI1 through ERAD. Further studies are needed to elucidate the regulatory mechanisms of the ubiquitinated BRI1 as an SGD1-bound substrate.

It is worth noting that the overexpression of the elite haplotype *SGD1^HI* increased panicle size, grain size, and 1000-grain weight per plant by 9.5, 12.2, and 10.3%, respectively (Fig. 7g–k and Supplementary Fig. 15), suggesting that *SGD1^HI* is a positive regulator of grain yield in foxtail millet. Moreover, *SGD1^HI*-overexpressing plants were more tolerant to foxtail millet blast disease (Supplementary Fig. 15). Remarkably, a recent study on *SGD1*'s rice ortholog *TT3.1* demonstrated that overexpressing the elite haplotype *TT3.1^CG14* in rice led to a twofold increase in grain yield per plant under heat stress[20]. Our experiments also showed that *SGD1* has a conserved function in grain yield regulation in rice, maize, and wheat (Fig. 2b–f and Fig. 3). Considering foxtail millet is known for its tolerance to environmental stresses and *SGD1^HI* is closely associated with high-yield performance under stresses, we suggested that *SGD1^HI* identified from foxtail millet holds significant potential for breeding high-yield and stress-tolerant crops. However, it is not clear whether the observed increase in yield is predominantly due to the overexpression of *SGD1* or the effects of the elite haplotype. Further experiments are needed, including overexpressing different SGD1 haplotypes in the same genetic background and constructing nearly isogenic lines. In addition, the molecular mechanisms of how *SGD1* maintains a relatively high grain yield under stress conditions are also intriguing. TT3.1 (OsSGD1) in rice is a thermosensor which transduces heat signals from PM to chloroplasts, protects thylakoids, and maintains photosystem[20]. SGD1 in foxtail millet can stabilize BR signaling (Fig. 6) and play a positive role in "protein processing in ER", "response to heat", and "photosynthesis" (Supplementary Fig. 15e, f). All these findings suggest that SGD1 may hold stable and high grain yield in crops via two mechanisms: (1) ER-localized SGD1 interacts with UBC32, modulates ER stress, and promotes BR-mediated growth. (2) PM-bound SGD1 can sense environmental stresses, including high temperature, transduce stress signals from the PM to chloroplasts or other organelles, activates downstream stress response/protective genes, and maintain photosynthesis efficiency in crops under stress. To sum up, *SGD1* may be an important hub gene that integrates stress responses and grain yield in crops.

In this study, we identified and characterized the SiUBC32-SGD1-SiBRI1 genetic module using foxtail millet. This is an important yet under-studied crop plant belong to the genus *Setaria*. Although the Setaria model has been proposed for years, very few reports that have used it as an experimental system. Our study showed that high-efficient forward gene cloning[21,50], transgenic engineering, mutagenesis, and protein functional analysis using foxtail millet protoplasts are all practicable, indicating that the Setaria model system is technically ready for exploring the mechanisms of specific genes. Advancements in positional cloning in maize and wheat are limited by the genomic complexity of these crops[51]. Our study has shown that *SGD1* is important for controlling grain yield not only in foxtail millet but also in maize and wheat. These demonstrate that *S. italica* can serve as an efficient model and has the potential to accelerate the discovery of new genes in major Poaceae crops.

## Methods
### Plant materials and growth conditions
The mutants *sgd1-1* and *sgd1-2* were isolated from a foxtail millet ethyl methanesulfonate (EMS) mutant library based on the Yugu1

variety with available reference genomes[52]. Both mutants were backcrossed to Yugu1 twice. All foxtail millet transgenic plants (*CR-sgd1*, *CR-Siubc32*, *sgd1/Siubc32*, Com-*SGD1*, Com-*ZmSGD1*, *OE-SGD1^H1*, and *OE-SiBRI1*) were obtained from the variety Ci846 (WT) which had high transgenic efficiency. Wheat transgenic plants (*CR-TaSGD1A*, *CR-TaSGD1B*, and *CR-TaSGD1D*) were obtained from the Fielder variety. Rice transgenic plants (*CR-OsSGD1*) were generated from the KitaaKe variety. For field experiments, all foxtail millet and rice plants were grown in the experimental field of the Institute of Crop Science, Chinese Academy of Agricultural Sciences (116.34′E, 39.97′N, Beijing, China), during the growth period (June to October) each year. Wheat (cv. Fielder) plants were grown at 22-24 °C under a 12-h light/12-h dark cycle in a greenhouse in Beijing (116.34′E, 39.97′N). For in-house experiments, foxtail millet plants were cultivated in a growth chamber under a 10-h light/14-h dark cycle at 30/26 °C, with a light intensity of 450–500 μmol m$^{-2}$ s$^{-1}$. Plants were photographed by a digital camera (EOS 500D, Canon, Japan). Florets, bristles, and grains were captured using a stereo microscope (M165FC, Leica, Germany).

### Phenotyping
Common agronomic traits such as plant height, panicle weight, panicle length, grain number per panicle, and 1000-grain weight were measured according to methods described in our previous study[36]. Three biological replicates were used in each treatment. Grain length and width were measured using grain analyzer software (SC-G Scanner, Wanshen Detection Technology Inc, Hangzhou, China). A total of 100–170 grains from each plant line were randomly selected for measurement. For the analysis of leaf architecture, proximal-distal distance (PDD), bending site length (BSL), and the full length (FL) of the second leaf from the top (Fig. 5c) was measured. Leaf drooping was quantified by the BSL to PDD ratio[19]. The angle of the third leaf was measured. For cell size measurements, foxtail millet florets were collected at the heading stage. Samples were fixed in FAA (formaldehyde:ethanol:acetic acid, 4:50:5, v/v/v) solution, dehydrated, embedded in resin, and cut into sections as described previously[53]. Mature seeds were also collected, cleaned, and dried in an oven (37 °C) for 3 days, and were examined under a scanning electron microscope (S3400N; Hitachi, Japan). Data and statistics were analyzed by GraphPad Prism8 (GraphPad Software, USA).

### Populations and gene mapping
The *sgd1-1* mutant was crossed with the foxtail millet variety SSR41[50] to obtain the *sgd1-1* × SSR41 F$_2$ mapping population. Approximately 761 homozygous recessive F$_2$ plants with a mutant phenotype were collected for DNA extraction and PCR mapping[50] with molecular markers (Supplementary Table 6). For BSA-seq, *sgd1-1* and *sgd1-2* mutants were backcrossed with Yugu1 to obtain the BC$_2$F$_2$ population. Leaf samples from 30-40 individual recessive plants from *sgd1-1* × Yugu1 and *sgd1-2* × Yugu1 BC$_2$F$_2$ populations were collected, and two DNA pools were constructed. The two DNA pools were then sent to Berry Genomics Company (Beijing, China) for whole-genome sequencing using a HiSeq 2500 platform (Illumina, USA). Clean reads were uploaded to China National Center for Bioinformation (CNCB) (https://ngdc.cncb.ac.cn/) under accession number CRA008001. Mutmap analyses were performed as described previously[54].

### Vector construction and plant transformation
For CRISPR/Cas9 genome editing, small-guide RNAs (sgRNAs) were designed according to the sequences of foxtail millet *SGD1*, *SiUBC32*, rice *OsSGD1*, and wheat *TaSGD1A/B/D* using CRISPRdirect (http://crispr.dbcls.jp/). The pYLCRISPR/Cas9-MH vector was digested with restriction enzymes to construct a sgRNA-containing plasmid as

described previously[55]. The primers used for CRIPSR/Cas9 vector construction are listed in Supplementary Table 6. For overexpression vector construction, the full-length coding sequences (CDS) of *SGD1* and *SiBRI1* were amplified from the cDNA of foxtail millet (cv. Yugu1). *ZmSGD1* was amplified from the cDNA of maize (cv. B73). These DNA fragments were cloned into the vector pCAMBIA1305.1-EGFP to generate *pUbi::SGD1-eGFP*, *pUbi::ZmSGD1-eGFP*, and *pUbi::SiBRI1-eGFP* vectors for overexpression. For the complementation test, a 9.3 kb genomic fragment (including promoter region and 3′UTR) of *SGD1* was amplified from Yugu1 and cloned into a modified pCAMBIA1305.1 vector. For analyzing the expression pattern of the *SGD1* gene, we cloned the 2.2 kb promoter region of *SGD1* into the pCAMBIA1305.1-GusPlus binary vector to generate a *pSGD1::GUS* construct. The primers used for vector construction are listed in Supplementary Table 6. All transgenic vectors were transferred to the Agrobacterium EHA105 strain, and the calli of foxtail millet (Ci846), rice (KitaaKe), and wheat (Fielder) were used for Agrobacterium-mediated transformation as described previously[56,57]. At least two independent transgenic lines for each vector were obtained and confirmed by sequencing and immunoblotting detection.

### Subcellular localization assay
To investigate the subcellular localization of SGD1, the CDS of *SGD1* was cloned into the pCAMBIA1305-eGFP vector, and then transferred to the p19 containing Agrobacterium strain GV3101 (pSoup-p19). The ER marker *35S::HDEL-mcherry* was co-expressed with *35S::SGD1-GFP* in *N. benthamiana* leaves by agroinfiltration. To test whether SiUBC32 colocalized with SGD1, the CDS of *SiUBC32* was inserted into the pBI121-mcherry vector. The *35S::SGD1-GFP* and *35S::SiUBC32-mcherry* vectors were co-transfected into foxtail millet protoplasts using PEG-mediated transfection[19]. Sections were imaged on a confocal microscope (LSM700, Zeiss, Germany). Images were analyzed and processed by ZEN Microscopy Software (Zeiss, Germany). The primers used for subcellular localization assay are listed in Supplementary Table 6.

### Yeast two-hybrid assays
Y2H cDNA library screening was performed using the DUALmembrane system (Dualsystems Biotech AG, Cat # P01001). The full-length CDS of *SGD1* was cloned into pBT3-SUC as bait and transformed into an NMY51 yeast strain with a NubG-fused cDNA prey library derived from young panicles. Positive clones that grew on synthetic defined media lacking leucine, tryptophan, histidine, and adenine (SD-LWHA) were selected for PCR and Sanger sequencing. To assess the protein interactions between SGD1, SiUBC32, and SiBRI1, the full-length or fragments of SiUBC32 and SiBRI1 were cloned into pPR3-N as prey. Each bait and prey construct, and empty vectors were paired up, transformed into yeast strain NMY51, and grown on synthetic defined media lacking leucine and tryptophan (SD-LW). After 2 to 3 days, positive clones were then shifted onto SD-LWHA to test interactions, and 3-AT (10 mM) was used to reduce background growth.

### Recombinant protein preparation and in vitro GST pull-down assay
For prokaryotic protein expression, the CDS of *SGD1* and *SiUBC32* were inserted into the gateway vector pCR™8/GW/TOPO and then cloned into gwpMAL-C2 and gwpGEX4T-1 to express MBP-SGD1 and GST-SiUBC32, respectively. The GST-SiBRI1c contained the kinase domain (KD) of SiBRI1 as described previously[19]. The constructs were expressed in the *E. coli* BL21 (DE3) strain. Proteins were purified using glutathione sepharose 4B beads (GE Healthcare, Cat # 17075601) and amylose agarose beads (New England Biolabs, Cat # E8035). Pull-down assays were performed as described previously in ref. 58, with modifications. GST (20 μL) and GST-SiUBC32 or GST-SiBRI1c agarose beads were washed with 1× PBS (137 mM NaCl, 2.7 mM KCl, 15 mM Na$_2$HPO$_4$, 4.4 mM KH$_2$PO$_4$) and preincubated with 20 μg bovine serum albumin

(BSA, Sigma, Cat # A7906) in 400 μL pull-down incubation buffer (20 mM Tris-HCl, pH 7.5, 100 mM NaCl, 0.1 mM EDTA, and 0.2% Triton X-100) for 30 min at 4 °C with gentle shaking. MBP or MBP-SGD1 protein were preincubated with 10 μL of prewashed glutathione agarose beads in 400 μL of pull-down incubation buffer for 30 min at 4 °C. The supernatant containing MBP or MBP-SGD1 was incubated with preincubated GST and GST-SiUBC32 or GST-SiBRI1c agarose beads for 1 h at 4 °C. Then, the protein–bead complex was washed with 1 mL of pull-down washing buffer (20 mM Tris-HCl, pH 7.5, 300 mM NaCl, 0.1 mM EDTA, and 0.5% Triton X-100) three times. Bound proteins were eluted with 2× SDS extraction buffer (125 mM Tris-HCl, pH 6.8, 2% β-mercaptoethanol, 4% SDS, 20% glycerol, and 0.25% bromophenol blue) and analyzed by immunoblotting with an anti-MBP antibody (TRANS, Cat # HT701). Western blot images were captured by a FUSION Solo S imaging system (Vilber, France).

### Split luciferase complementation assay
The full-length CDS of *SiUBC32*, *SiBRI1* or *SiSGD1* were cloned into the pCAMBIA-1300-cLUC or pCAMBIA-1300-nLUC binary vectors, to generate N-terminus (nLUC) or C-terminus (cLUC) fused and truncated luciferase tags on the target proteins respectively. All constructs were paired and transformed into Agrobacterium strain GV3101 (pSoupp19) for agroinfiltration of *N. benthamiana* leaves. After 36–48 h of co-infiltration, the luciferase substrate D-luciferin (Promega, Cat # P1043) was sprayed onto the leaf surface. The chemiluminescence signal was observed using the NightSHADE LB 985 plant imaging system (Berthold Technologies, Germany).

### Evaluation of plant growth under BR treatment
For evaluating primary root growth inhibition in the presence of 24-epi-Brassinolide (eBL, Sigma, Cat # E1641), Ci846 and *sgd1* seeds were sterilized and used for eBL treatment according to our previous study[28]. Uniformly grown Ci846 and *sgd1* seedlings were then transferred to 1/2 MS medium with or without eBL (0.01 μM or 0.1 μM). Root length was imaged and measured after 5 days of continuous growth. For leaf drooping measurement, Ci846 and *sgd1* seedlings were grown in plastic pots containing potting soil, peat moss, and vermiculite (2:1:1, v/v/v). Seedlings grown at a four-leaf stage were sprayed with 5 μM eBL or mock for 3 days. Leaf drooping and leaf angle measurements were described in the subsection *Phenotyping*.

### Co-immunoprecipitation (Co-IP) assay
The full-length CDS of *SiUBC32*, *E2CK*, *SGD1*, and *BRI1* were amplified by PCR from foxtail millet (cv. Yugu1) cDNA and cloned into pHBT plant expression vectors. These vectors were fused to an HA, FLAG, or MYC tag. Co-IP assays were performed as described previously in ref. 59. Yugu1 protoplasts were transfected and incubated for 10 h. Total protein was extracted from protoplasts using the protein extraction buffer containing 1 mM EDTA, 0.5% Triton X-100, and a protease inhibitor cocktail (Roche, Cat # 11697498001). The samples were centrifuged at 12,000×*g* at 4 °C for 15 min. The supernatant was incubated with anti-FLAG (Sigma, cat # F1804) antibodies at 4 °C for 2 h with gentle shaking. Then protein-G-agarose beads (Thermo Scientific, Cat # 20397) were added and the incubation was continued for another 2 h. The agarose beads were collected by centrifugation (100×*g*) at room temperature for 2 min, washed four times with washing buffer containing 1 mM EDTA and 0.1% Triton X-100, and washed once with 50 mM Tris-HCl at pH 7.5. The immunoprecipitated proteins were detected by immunoblotting with indicated antibodies.

### In vitro ubiquitination assay
The full-length CDS of *SiUBC32* and *E2CK* were cloned into the pACYCDuet-S vector. Truncated SGD1 (SGD1c) and SGD1c with site mutations (C426A and H443A) in the RING domain (mSGD1c) were cloned into the pACYCDuet-Myc vector. Truncated SiBRI1 (SiBRI1c)

was inserted into the pCDFDuet-MBP vector. The vectors containing E1 (*pCDFDuet--HA-UBA1-S*) and ubiquitin (*pET28a-FLAG-Ub*)[25] were grouped and co-transformed into *E. coli* BL21 (DE3) competent cells. Proteins from co-transformed BL21 (DE3) bacteria were expressed. The ubiquitination assay using the bacterial reconstituted system was performed as previously described in ref. 25.

### Measurement of BRI1 protein stability
The specificity of the SiBRI1 antibody was identified. The SiBRI1 amino acid sequence was blasted against the foxtail millet protein database (Phytozome, *Setaria italica* v2.2). The result showed that SiBRI1.L1, SiBRI1.L2, and SiBRI1.L3 were highly similar to SiBRI1 (Supplementary Fig. 10). The coding sequences of *SiBRI1*, *SiBRI1.L1*, *SiBRI1.L2*, and *SiBRI1.L3* were cloned into the pCAMBIA1305-eGFP vector and transferred into foxtail millet protoplasts. After 10 h incubating, total protein was extracted and immunoblotted using anti-SiBRI1, anti-GFP, and anti-Actin antibodies. To measure SiBRI1 abundance in different mutants, 14-day-old Ci846, *CR-sgd1*-L1, *CR-sgd1*-L2, Yugu1, *sgd1-1*, and *sgd1-2* seedling leaves were used for immunoblotting with anti-SiBRI1 and anti-Actin antibodies. For CHX treatment, Ci846, *CR-sgd1*, and *OE-SGD1* seedlings were grown in plastic pots containing potting soil, peat moss, and vermiculite (2:1:1, v/v/v). Then, 14-day-old seedling leaves were treated with 100 μM CHX (Cycloheximide, MCE, HY-12320) for 0, 3, or 6 h. Total proteins were detected by immunoblotting with anti-SiBRI1 and anti-Actin antibodies. Western blot images were analyzed by ImageJ (https://imagej.net/downloads, Version 1.49). The uncropped and unprocessed scans of all blots were provided in the Source Data file.

### Transcriptome sequencing and qPCR
In total, five different kinds of plants, i.e., seedlings of wild-type Yugu1 plants, *sgd1-1* mutant plants at the four-leaf stage, young panicles of Ci846, *CR-sgd1*-L1, and *OE-SGD1^H1*-L1 at the heading stage, were collected for RNA-seq analysis. Fifteen RNA-seq samples (three biological replicates for each stage) were sent to Berry Genomic Company (Beijing, China) for RNA extraction and transcriptome sequencing. Fifteen cDNA libraries were constructed and sequenced on a HiSeq 2500 platform in 150-bp paired-end mode (Illumina, USA). Sequencing data were submitted to the CNCB public database as stated in the following accession number part. Clean reads from each library were mapped and analyzed as described previously in ref. 60. The relative expression of six genes involved in BR biosynthesis or BR-regulated pathways in WT and mutant plants was measured by qPCR. Total RNA was extracted using TRIzol (Thermo Fisher Scientific, Cat # 15596026). RNA (5 μg) was reverse-transcribed into cDNA using the PrimeScript™ II 1st Strand cDNA Synthesis Kit (Takara, Cat # 6210 A). qPCR was performed using FS Universal SYBR Green Master (Roche, Cat # 4913914001) with gene-specific primers (Supplementary Table 6). Foxtail millet *SiActin* and *SiCullin* were used as reference genes.

### Selective sweep and haplotype analysis
A total of 1681 *Setaria* germplasms (530 wild species, 694 landraces, and 457 cultivars) were used for whole-genome selective sweep analysis. Among them, 942 germplasms were collected and sequenced in our precious study[36], and other high-throughput sequencing data were obtained from published databases[35,37]. Individuals with inbreeding coefficients below 0.8 and markers with heterozygosity rate over 0.2, minor allele frequency below 0.05, and missing rate over 0.1 were excluded using VCFtools (http://vcftools.sourceforge.net/). *Fst* was calculated in VCFtools with a 200-kb window set, and the nucleotide diversity value (*π*) was calculated in VCFtools with a 2-kb window set in the target regions 7.2 Mb and 8.0 Mb. In the gene region, *π* was calculated using PopGenome in R (https://github.com/tonig-evo/workshop-popgenome) with a window size of five SNPs and a step of

two SNPs. The top 5 and 1% were used as thresholds to screen the loci involved in domestication and improvement, respectively. A total of 960 *Setaria* varieties were grown in the experimental field (116.34'E, 39.97'N, Beijing, China) from June to October 2018. Thirteen agronomic traits were investigated as described in the subsection *Phenotyping*. Haplotype analysis was carried out using an in-house R script (https://gitee.com/zhangrenl/genehapr).

## Statistics and reproducibility

General information on study design, statistical analysis of data and reproducibility of experiments were stated in related figure/table legends.

## Reporting summary

Further information on research design is available in the Nature Portfolio Reporting Summary linked to this article.

## Data availability

All data needed to evaluate the conclusions in the paper are present in the paper and/or the Supplementary Information. The high-throughput sequencing data generated in this study were deposited into China National Center for Bioinformation CNCB with accession numbers CRA007999 and CRA008001 which are publicly accessible. Gene sequence information of foxtail millet, maize, rice, and *Arabidopsis* from this study can be found in Phytozome v13 (*Setaria italica* v2.2, *Zea mays* RefGen_V4, *Oryza sativa* v7.0), TAIR, or NCBI, under the following accession numbers: *SGD1* (Seita.9G123200), *SiUBC32* (Seita.9G428900), *SiE2CK* (Seita.9G236200), *SiBZR1* (Seita.2G367800), *SiBAS1* (Seita.5G123900), *SiBIN2* (Seita.5G145300), *SiCullin* (Seita.3G037700), *SiD2* (Seita.5G139200), *SiCYP51G3* (Seita.2G356300), *SiGLR2.7* (Seita.1G009400), *SiCBF2* (Seita.2G280200), *SiBRH1* (Seita.7G209400), *SiBRI1* (NCBI Gene ID: LOC101765569), *SiBRI1.L1* (Seita.9G296000), *SiBRI1.L2* (Seita.2G165600), and *SiBRI1.L3* (Seita.6G117300), *ZmSGD1* (Zm00001d013466), *OsSGD1* (TT3.1, Os03g49900), and *UBA1* (AT2G30110). Gene sequence information of wheat is available at Ensembl Plants with accession numbers: *TaSGD1A* (TraesCS4A02G271200), *TaSGD1B* (TraesCS4B02G042900), and *TaSGD1D* (TraesCS4D02G040200). Source data are provided as a Source Data file along with this paper. Source data are provided with this paper.

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

## Acknowledgements

We are grateful to Professor Yunhai Li for his technical assistance, Professor Dongping Lu for providing the plant ubiquitination system in bacteria, and Dr. William Teale for editing the English text. This work was supported by grants from the National Key R&D Program of China (2021YFF1000100/2021YFF1000103 and 2019YFD1000700/2019YFD1000701), the National Natural Science Foundation of China (32241042, 32201801, and 31800222), China Agricultural Research System (CARS-06-13.5), Agricultural Science and Technology Innovation Program of the Chinese Academy of Agricultural Sciences, Hebei Science Fund for Distinguished Young Scholars (C2021503001 to M.Z.) and China Postdoctoral Science Foundation (2022M713434 to Z.Z.).

## Author contributions

X.D. and S.T. designed the research. S.T., Z.Z., and X.L. performed most of the experiments with the help of Y.S., D.Z., Y.G., H.Zhang, L.Z., Y.W., M.Z., and W.Z. H.Zhi, S.T., D.Z., Y.G., and H.Zhang. performed the field experiments. K.W. and X.Y. performed transgenic experiments on wheat. D.L., Q.H., and R.Z. analyzed bioinformatic data. S.T., Z.Z., X.L., L.Z., and Y.W. analyzed experimental data and wrote the manuscript. X.D., G.J., W.T., and C.W. discussed and revised the manuscript.

## Competing interests

The authors declare no competing interests.
