## [Peer Review File · Nature Communications]

An E2-E3 pair contributes to seed size control in grain cropsREVIEWER COMMENTS

Reviewer #1 (Remarks to the Author):

General comments

This manuscript describes a mechanism that contributes to cell size and grain yield in foxtail millet (*Setaria*). This is an important yet under-studied crop plant. There are very few reports that have used *Setaria* as an experimental system, especially to define a new ubiquitylation mechanism that stabilises a brassinosteroid receptor. Therefore, the work is technically demanding compared to eg rice, and it is also comprehensive, going from gene discovery through a mutant screen to mechanism and through to characterising haplotypes in breeding lines that improve yield. The technical quality is very high and is of general interest in that it demonstrates the power of modern plant science to identify genes and mechanisms underlying key traits. However, this technical expertise is not matched by the quality of writing and explanation, which is so low in places it is hard to understand what the authors are trying to say. Also, the ms is much too long, which detracts from its scientific importance.

Specific comments

The Introduction is far too long and poorly written, can shorten 1st para, second para is a list with no context to the subject. The Ub system description is duplicated in the discussion so just have it in the Discussion.

BR para is important in the Introduction but again it is still too long

Results

L120-122. I did not understand these sentences about absence of tillers and grain yield. Could delete them.

L 129. It is implicit that the differences seed were statistically significant, so this sentence can be deleted.

L140-152. This paragraph can be shortened substantially.

L170 "CRIPSER-edited"?

L263 to claim the interactions are synergistic the authors need to describe in the text here the extent to which the interactions are more than additive

L270 "which calls for the characterised role " meaning not clear.

L275 explain using references to past work why these phenotypes were selected. Also, 5 uM eBL is a very high concentration- please check.

L 295 "tuned to" makes no sense

Discussion

It would be interested to discuss the ER localisation of SGD1 and how this location may be important for BRI1 trafficking. There is much know about this that the authors don't touch on,. he focus on quality control is sort of OK but it doesn't explain higher levels of functional BRI1. The Discussion could also make more of the use of SGD1 haplotypes in crop improvement instead of general comments about the *Setaria* system.

Methods

These are thoroughly described

Figures.

These are very well presented.

Figure 1 shows grain number per panicle and 1000-grain weight as a key yield trait. In Figure 2 only grain size is used for the rice and wheat analyses. It is important to include grain number per panicle in Figure 2 to show the effect of SGD on overall yield.

Figure 3. There needs to be an optical cross-section of FM4-64 and GFP signals to show co-location.

Figure 6 panel H should include a *bri1* ko control to show the specificity of the assays of BRI1 levels using SiBRI1 antibodies. Also, relative BRI1 levels could be quantified by scanning.

Reviewer #2 (Remarks to the Author):

Overall, a strong study. It adds to an emerging story that the ubiquitin pathway impacts fundamental processes influencing agronomic traits. Not surprising result given the body of work in general that has emerged implicating the ub pathway is just about every process! Here the work is strong is demonstrating that SGD1 in *Setaria* is responsible for the phenotype, demonstrating that the related gene in other grasses works similarly. The significance of this effect is strengthened by creating mutants in wheat and rice, demonstrating its conservation. The effect of loss of SGD1 is supported by multiple mutants, and complementation with the coding region. Its biochemical function as an E3 ligase is confirmed.

Other strengths are identifying an interacting E2 and showing genetically its involvement in the same process. An impactful aspect is the allelic variation that can quantitatively influence the phenotype. Finally, the authors showed that SGD1 interacts with and ubiquitinates the brassinosteroid receptor, BRI1. The effect of this modification is not the canonical facilitation of degradation, but rather the protein is present at higher levels in presence of SGD1. This work will be of interest to both basic and applied researchers.

Despite many strengths, I do have several concerns. First, the authors seem overly concerned with identifying SGD orthologs (as opposed to related- homologs) in other species. The work in identifying functional orthologs in grasses is supported by their experimental evidence (mutational analyses through KOs in other species), but in dicots the identity of orthologous protein is much more tenuous, and I think should not be stressed. The authors try to extend the orthology comparisons to subcellular localization and, in part because their intracellular localization data are weak, the comparison seems forced and without sufficient support. Second, the subcellular localization data are not convincing, but they are not critical to the main impactful results, so could be removed without affecting the conclusions. They do support the overall model that BRI1 abundance is affected in the ER or PM. The conclusion that SGD1 affects BRI1 stability could have stronger evidence. Third, there are multiple instances of incorrect or overinterpretation of the data. See below for more complete description/specifics of these concerns.

1. Lines 82-102. This paragraph in the introduction describes BR signaling and the current knowledge of the intersection between ubiquitin pathway and BR pathway. In addition, it describes two BR-related mutants in *Setaria*. I think the order of topics within the paragraph is backwards. To aid the general reader, I suggest that, after the first sentence (revised as suggested below in 'Minor' section), the authors start with a description of BR signaling in general (using already existing sentences, starting at end of line 90). In other words, start the

paragraph with the general BR response pathway, lines 90-102. I don't think the descriptions of the 2 *Setaria* BR mutants are relevant in the introduction and should be removed. The mutant *dpy1* used in the work is mentioned in the appropriate section (but should have a citation) when utilized in relation to testing the role of BR signaling.

2. Lines 100-102. What is the evidence that "there should be other E3 ligases that directly regulate either the protein stability, activity, or localization of the BRI1 receptor..."? I think the reader should be provided with some evidence to substantiate this claim that there should be other E3 that regulate BRI1 receptor function. This evidence could come from mutational studies, for example. For example, what is the regulation of BRI1 in a *pub12/13* double mutant background? If it is still degraded, then this result indicates that additional activity also regulates BRI1 abundance. I think the rationale for the current study would be strengthened by providing some experimental support for the hypothesis that there should be other E3s regulating BRI1.

3. Results, lines 167-170. The authors write, "Phylogenetic analysis suggested that SGD1 is an ortholog of Arabidopsis PPRT1 and rice TT3.1 gene (Supplementary Figure S3), which were reported to be involved in salt and high temperature stress responses."

Supp. Fig S3 does not provide evidence of orthology to these specific proteins. This is because only proteins with shared domains are included, so a greater identity to these proteins over other E3 is not shown. There are 3 Arabidopsis proteins in the figure, which is PPRT1? Next, which of the 3 Arabidopsis proteins is most similar to SGD1? They all branch off together, so I don't see how you can say one of these 3 is the ortholog. Could the authors clarify?

There is only one rice protein in the analysis, so we can't evaluate orthology. A statement that TT3 is the only protein with these 2 domains and has the highest identity to SGD1 over all other rice proteins are evidence to support orthology.

Because of the uncertainty, making direct comparisons of SGD1 to PPRT1 localization and function seems problematic.

Lines 210-212. Related to protein relationships, the authors did not provide us with information in support of their conclusion that *Setaria*.9G428900 is the ortholog of UBC32. From sequence comparisons, it is most similar UBC32

4. The results from the intracellular localization experiments are weak (Fig 3C,D). The lack of a merged color in 3C suggests that the fusion protein is NOT localized to the PM (or the FM signal is overwhelming the GFP signal). Cells with more evident cytoplasm should be used to assign localization more definitively.

5. Lines 249-264 and Figure 4. The authors generate a double *ubc32 sgd1* mutant and conclude that the two genes act synergistically to regulate grain yield. I don't see this synergism. It is difficult to evaluate from the text because we are not given values for the single mutants (please include the numbers for the single mutants), but looking at the graphs, the double mutant is more severe than each single, but the effect appears additive, not synergistic. Could the authors provide more support for their synergistic hypothesis?

It would also be of value to provide a model. If they both work together and are required together for ubiquitination of BRI1, then I would expect the double mutant to look like each of the single mutants, not to have a greater effect.

6. Lines 282-285. The following statement seems incorrect in concluding that *sgd1* is insensitive to BL. The authors write, "A clear inhibition effect was observed at 0.01 μ M and

0.1 μ M eBL treatment in WT seedlings, whereas the corresponding effects were not seen in the *sgd1* mutants (Figure 5F and G). These results indicated that *sgd1* is insensitive to BR.”

In the panel (G), application of eBL does result in shorter roots in the *sgd1* mutant, but the effect is not as strong as in wild type roots. This result indicates that the roots are not insensitive, but less sensitive (hyposensitive) to exogenous BL. Authors please correct. (the overall conclusion that *sgd1* has reduced response to BL – lines 308-310 – is well supported by the data.

7. Figure 6F. From the data in this panel, the authors conclude that “SiBRI1 ubiquitination and protein stability were reduced in *sgd1* compared to WT plants” (statement in 6F legend). The figure only shows reduced ubiquitination, not stability. The experiment immunoprecipitates ubiquitinated proteins in total, then visualizes in that IP pool tagged-BRI1. They can only assess the level of ubiquitinated forms.

8. Discussion, lines 443-449. The authors give a list of reasons in support of their conclusion that “interaction and ubiquitination process led to enhancement of BRI1 accumulation and BR signaling rather than degradation of BRI1 for these following reasons.” The first two reasons listed are not in support of this conclusion. They support the hypothesis that SGD1 ubiquitinates BRI1 in vivo, not that SGD1 leads to enhancement of BRI1 accumulation and BR signaling.

Evidence that SGD1 enhances BRI1 accumulation and BR signaling are the observed hyposensitivity of *sgd1* mutants to BR, that levels of BRI1 appear to be higher when SGD1 is over-expressed, that ubiquitination is lower in *sgd1* mutants and BZR1 phosphorylation status is affected (these arguments are included further down in the discussion). These first statements should be removed or included in a prior conclusion that BRI1 is an in vivo substrate of SGD1.

9. The main claim mechanistically is that SGD1 ubiquitinates and stabilizes BRI1. This is supported by some data that are not as conclusive as one would like. While there are several experiments consistent with this interpretation, there is no direct measurement of BRI1 protein stability. Given that they have Ab to the *Setaria* BRI1, the authors could immunoblot for BRI1 in WT and in multiple *sgd1* mutant lines to compare BRI1 abundance. Important to demonstrate the effect in all the mutant lines. Second, the authors should show that co-expression of the catalytically inactive form of SGD1 does not affect BRI1 accumulation (add to experiment shown in Fig 6G). The authors should perform a cycloheximide experiment- to measure stability by monitoring protein abundance over time in the presence of cycloheximide. It should be possible with plant roots (for example since immunoblotting extracts yields signal-see Fig 6H). Or the same protocol could be done using protoplasts after transfection --- incubate for expression – then add cycloheximide and monitor BRI1 abundance over time +/- coexpression with SGD1. (Figure 6G experiment with a cycloheximide time course).

Minor. Typos. use of wrong word

1. Line 58-59, I think there is a typo- “is one of the examples that extent knowledge” should be “is one of the examples that extends knowledge”?
2. Line 74-75 modify, “that work redundantly in controlled seed size through regulating cell...” to “that work redundantly in controlling seed size through regulating cell...”
3. Lines 79-91, not sure of meaning of this sentence: “Beyond expectation, as E3’s working partner, few E2 was identified to be involved in grain yields regulation, let alone any E2/E3 pairs in plants.” I think that this statement, if I think I understand its meaning, should just be deleted. E2/E3 pairs have been reported in plants, and what does “few E2” mean? I don’t

think this statement provides a strong argument in support of the significance of the work reported here and should just be removed.

4. Line 83, “arrange” should be “range”. The sentence containing “arrange” is awkward even with correction. How about “Brassinosteroid (BR) is a phytohormone that broadly regulates plant growth and development, as well as stress responses.” All phytohormones are important, so I don’t think that term should be included.

5. line 86, change “homolog” to “homologous”

6. lines 106-107, revise “Homologous comparison suggested” to “Homology comparisons suggest”

7. Line 114, modify “genetical” to “genetic”

8. Line 115, modify “support the” to “support that the”

9. Line 116, modify “grain yields” to “grain yields”

10. Line 116-117, modify “high-yields breeding” to “high-yield breeding”

11. Figure S1. Could the gene names be rotated so they are as right-side-up as possible? Hard to read the figure when many are upside down.

12. Figure S1. What is the species represented by EI?

13. Figure S1. Could the authors include in the legend the abbreviations used for each species- in parenthesis after the name- for example, *Zea mays*, Zm...

14. Line 168, could the AGI number for PPRT1 be given in the text? Because Fig S1 only shows AGI numbers for the Arabidopsis proteins so the reader can’t know which protein is PPRT1.

15. Line 170, “CRIPSER” should be “CRISPR”

16. Line 171, “ossgd1” should be “Ossgd1”. There is more than one crispr line, so the text should reflect that information-such as, “ two CRISPR-edited rice Ossgd1 lines both showed small grains (Figure 2G and H).”

17. Line 172 “conservative” should be “conserved”

18. Line 173, “firstly isolated two SGD1 homologs” change to “first identified two SGD1 homologs”

19. Line 174, Not sure what is meant by “For its higher DNA similarity,” Was this one chosen because its protein (not DNA) sequence had highest identity to SGD1? Please clarify.

20. Line 191, “tissues” should be replaced with “organs”. The parts of the plant tested are different organs.

21. Line 197, “fusing” should be changed to “fusion”

22. Line 202, I don’t see yellow arrows in Fig. 3C. Maybe in reducing their size, the color differential between white and yellow is minimized? I suggest different shapes instead of different colors, one using an arrowhead only, another using an arrow.

23. Line 211. Modify “which is a kind of ubiquitin-conjugating enzyme” to “which is a ubiquitin-conjugating enzyme”

24. Line 223, Modify “SiiUBC32” to “SiUBC32”

25. For many lines throughout the text, the authors should capitalize the S of “siubc32” and in all other mutant alleles because the Si of the name refers to the plant species and is not lowercase even in mutant allele names.

26. line 265, heading title, suggest modifying “Loss of function in SGD1 is involved in BR signaling” to “SGD1 is involved in BR signaling”. The loss-of-function mutant in SGD1 allows you to discover that SGD1 has a role in BR signaling.

27. Line 273, Not sure what is meant by “isolated” ? Maybe could say, a naturally occurring active brassinolid? Just a suggestion, could the authors clarify?

28. Line 276, “while the Leaf angle” should be “while the leaf angle”

29. Line 287, “-taget” to “-target”

30. Line 288, “Chang” to “Change”

31. Line 295, “tuned” to “turned”

32. Lines 301-302. I think the authors mean the SGD1 plays a role in the negative feedback of BL biosynthesis, not the loss-of-function version, because the mutant has higher levels of mRNA for BL biosynthetic genes.

33. Line 357, modify “BR-insensitive phenotypes” to “BR-hypo-sensitive phenotypes”

Reviewer #3 (Remarks to the Author):

The authors present an extremely thorough phenotypic and genetic analysis of an E3 ligase that influences grain yield in *Setaria italica* including mutant analysis, identifying interacting partners and investigating the role of brassinosteroids on the phenotype. They demonstrated that SGD1 orthologs in wheat and rice influence grain yield and that a maize ortholog can rescue the *Setaria* mutant. Thus, SGD1 is of broad importance in grain yield. They also conducted a population analysis that indicated the SGD1 locus was under selection during domestication and identified 3 haplotypes associated with domestication, including one that is dominant in elite cultivars. This haplotype is also associated with the greatest increase in grain yield. Overall, the experiments conducted were very thorough and supported the conclusions. While the figures are rather complex, they present the data in a comprehensible fashion and show numerous lines of evidence to support their conclusions. I think the extremely broad characterization spanning from protein and molecular characterization to other grasses and population analysis combined with the apparent importance of the gene in grain yield and domestication will make this work of great interest to readers.

I have the following suggestions to improve the manuscript.

The manuscript needs to be carefully edited for grammar and language because there are many awkward sentences. I note a few below, but my list is not comprehensive.

The authors write that SGD1 and SiUBC32 are primary partners in an E2-E3 complex. However, their double mutant analysis showed that they act additively. This indicates that they are not simply acting in the same linear pathway or as a single complex. The authors should discuss this in the results and discussion (line 435) and perhaps postulate possible reasons for this interesting genetic observation and the fact that they clearly interact based on other evidence. I also noticed that the SGD1 band is much weaker than the SiUBC32 band in their pulldown experiments. That may be something they could postulate about.

The interpretation of the results of the overexpression of SGD1 in cultivar in Ci846 is not straightforward because the gene inserted is a different haplotype than the native SGD1 gene in Ci846. Since the haplotype inserted is associated with higher yield than the native Ci846 haplotype, it cannot be concluded that the observed increase in yield is due to overexpression rather than to the haplotype. The authors should acknowledge this in the discussion and abstract.

The authors make numerous mentions of the *Setaria* model system. This may be confusing to many readers who are familiar with *Setaria viridis* as a model plant and *Setaria italica* as a grain crop. It would be clearer if the authors include an introductory sentence indicating that they are using foxtail millet, *Setaria italica*, as a model system for grains in general.

Line 73: They should add references for this sentence "A few ubiquitin-related proteins involved in grain yield control were identified."

Awkward sentences need to be edited on lines 53, 58, 69, 79, 121, 172, 205-206, 253, 268, 270, 297, 375, 377, 393-394, 431, 453, 520, 696-697

Line 83 change arrange to array

Line 120 add "The" before "majority"

Line 197 change fusing to fusion

Line 186 reword. SGD1 does not have a new function

Line 278 change bended to bent

Line 380 change suffered to underwent

Line 432-433 cite the E2 ligases reported to regulate grain yield.

Line 458 not clear what is meant by quality control

Line 541 change trait to traits

Line 707 change grown to grew

Responses to Reviewers

Reviewer #1 (Remarks to the Author):

General comments

This manuscript describes a mechanism that contributes to cell size and grain yield in foxtail millet (Setaria). This is an important yet under-studied crop plant. There are very few reports that have used Setaria as an experimental system, especially to define a new ubiquitylation mechanism that stabilises a brassinosteroid receptor. Therefore, the work is technically demanding compared to eg rice, and it is also comprehensive, going from gene discovery through a mutant screen to mechanism and through to characterising haplotypes in breeding lines that improve yield. The technical quality is very high and is of general interest in that it demonstrates the power of modern plant science to identify genes and mechanisms underlying key traits. However, this technical expertise is not matched by the quality of writing and explanation, which is so low in places it is hard to understand what the authors are trying to say. Also, the ms is much too long, which detracts from its scientific importance.

Response to general comments

Thank you very much for critically reviewing our manuscript. Your positive comments on the experimental design, technical quality, and advantages of the Setaria model system are really encouraging. We acknowledge our shortcomings in English writing and thus made concerted efforts to improve language and clarity by reorganizing and simplifying content according to your suggestions. We also added data and experiments to address your concerns. Please refer to the point-by-point responses for further details.

Specific comments

Comment 1: *The Introduction is far too long and poorly written, can shorten 1st para, second para is a list with no context to the subject. The Ub system description is duplicated in the discussion so just have it in the Discussion.*

Response 1: We agree that the Introduction is long and needs improvement. According to your suggestion, we removed duplicated content and data weakly related to the subject. In addition, the second paragraph was rewritten (lines 60-70), and the introduction section was shortened by approximately 15%.

Comment 2: *BR para is important in the Introduction but again it is still too long*

Response 2: Thank you, we shorten this paragraph. In addition, another reviewer (the reviewer 2, comment 1 and 2) also pointed out this paragraph is important and gave some specific directions. Combine your and the reviewer 2's suggestion, we re-organized this part (lines 71-87).

Comment 3: *Results. L120-122. I did not understand these sentences about absence of tillers and grain yield. Could delete them.*

Response 3: These sentences were deleted because they were weakly related to the context.

Comment 4: L 129. *It is implicit that the differences seed were statistically significant, so this sentence can be deleted.*

Response 4: The sentence was deleted as requested.

Comment 5: L140-152. *This paragraph can be shortened substantially.*

Response 5: We agree that the positional cloning of the candidate gene does not need to be described in the main text. We shortened this paragraph (lines 120-127) and moved information to the legend of Fig. 2a and the footnote of Supplementary Table 2.

Comment 6: L170 “CRIPSER-edited”?

Response 6: Sorry for the typo. We revised to CRISPR (lines 168).

Comment 7: L263 *to claim the interactions are synergistic the authors need to describe in the text here the extent to which the interactions are more than additive*

Response 7: Thank you for the valuable suggestion. We misused the word “synergistic” when describing the relationship between SGD1 and UBC32. To solve this problem, we deleted “synergistic” throughout the manuscript, added some data and descriptions. 1. We added a supplementary table that record values of the agronomic traits of WT, *sgd1*, *Siubc32*, and *Siubc32/sgd1* (revised Supplementary Table 8). From this table, it is clear that the growth inhibition observed in the double mutant is more severe compared to each of the single mutants in all the investigated traits. We further compared the reductions observed in two single mutants with that in the double mutant. We found that the reductions of plant height, flag leaf length, and panicle length in the double mutant are much greater than each single mutant, suggesting these two genes acts additively in regulating these three traits. However, for grain yield-related traits, including 1000-grain weight, grain length, and grain width, the double mutant was slightly more severe than each single mutant. Take 1000-grain weight for example, *sgd1* showed a 40.0% decline, *Siubc32* exhibited a 40.5% reduction, and the double mutant showed a 43.2% reduction, which is only slightly greater than each single mutant. This result suggests that SGD1 and SiUBC32 have partial overlapping functions in regulating grain yield. 2. At the biochemical level, SGD1 and SiUBC32 are partners in an E2-E3 complex, which further modify their target proteins such as BRI1. However, we should recognize that SGD1 and SiUBC32 are not simply acting in the same linear pathway or as a single complex. There are other E3s that can interact with UBC32 (Chen et al., 2021). Similarly, there are also other E2s can form a complex with SGD1 (Zhang et al., 2022). Connect with phenotypes of *sgd1*, *Siubc32* single and double mutants in grain-yield related traits, we suggest that SGD1 acts, at least in part, in a common genetic pathway with SiUBC32 to control grain-yield related traits. Related content was added in revised manuscript (lines 236-256).

Comment 8: L270 *“which calls for the characterised role “meaning not clear.* L275 explain using references to past work why these phenotypes were selected.

Response 8: Thank you for indicating the lack of clarity. We clarified why bristle length and leaf droopiness were selected.

“In *Setaria*, two distinctive BR-related biological processes were reported including bristle development (Yang et al., 2018) and leaf droopiness (Zhao et al., 2020). Notably, *sgd1* mutant exhibited short bristle length (Fig. 5a-b) and compact leaf architecture (Supplementary Fig. 9), which resembled the BR-defective phenotypes reported in previous studies. We therefore considered that *SGD1* may be involved in the BR-related pathway.” (Lines 258-262).

Comment 9: *Also, 5 μM eBL is a very high concentration- please check.*

Response 9: Yes, we confirmed that 5 μM eBL is the correct concentration. We sprayed 5 μM eBL on foxtail millet leaf blades and measured leaf droopiness, as described previously (Zhao et al., 2020). We can easily detect differences in leaf droopiness in foxtail millet using this eBL concentration. Nonetheless, the eBL concentration used in rice lamina joints is much lower (10-1000 nM) (Tian et al., 2021). The optimal concentration of eBL depends on plant species (rice or foxtail millet), tissues (lamina joint and leaf blade), phenotypes (leaf angle and leaf droopiness), and methods of treatment (dripping and spraying).

Comment 10: *L 295 “tuned to” makes no sense*

Response 10: Sorry, it is a misspelled word here. We corrected it to “turned into” (line 284).

Comment 11: *Discussion. It would be interested to discuss the ER localization of SGD1 and how this location may be important for BRI1 trafficking. There is much known about this that the authors don't touch on, he focus on quality control is sort of OK but it doesn't explain higher levels of functional BRI1.*

Response 11: Thank you for this professional suggestion. Your idea inspired us to consider BRI1 trafficking rather than the quality control of BRI1 in the ER. This a very interesting topic, we believe that two aspects of BRI1 trafficking may require SGD1. First, BRI1 is synthesized in the ER and transported to the Golgi apparatus for further modification and sorting (Jaillais et al., 2016). These processes involve membrane-bound organelles, including the ER, Golgi apparatus, and the trans-Golgi network/early endosome, and the ubiquitination of the target proteins (Martins et al., 2015). In our study, we observed a significant increase in BRI1-GFP intracellular fluorescence signals compared to that of the PM in the root of BRI1-GFP/*sgd1* plants (Supplementary Fig. 16). In addition, we found that several SNAREs (soluble N-ethylmaleimide sensitive factor attachment protein receptors) that are important for protein trafficking, such as SYP61 and VAMP726, were included in the putative SGD1-interacting proteins identified by Y2H screening (Supplementary Table 7). In consideration of the report that SNAREs mediate the BRI1 trafficking to PM (Zhang et al., 2019), we speculate that SGD1 may work with SNAREs to ensure the proper trafficking of BRI1s from the ER to the PM, where they can execute their BR-receptor function. On the other hand, PM-resident BRI1 undergo internalization through the endocytic pathway (Martins et al., 2015). This process is also regulated by ubiquitination (Martins et al., 2015; Zhou et al., 2018) Internalized BRI1 may be recycled to the plasma membrane or to the late endosomes/multivesicular bodies (MVB)

for eventual degradation in the vacuole. A few ubiquitination sites of BRI1 have been identified as being important for its degradative sorting to the vacuole (Martins et al., 2015). Two E3 ligases PUB12/13 which can mediate BRI1 polyubiquitination and promote BRI1 endocytosis and degradation were also found. In *Arabidopsis pub12/13* mutant, the abundance of BRI1 protein and its residence time at the plasma membrane are increased, while the endosomal pool of BRI1 is reduced (Zhou et al., 2018). In our study, SGD1 ubiquitinates and stabilizes BRI1, and a significant increase of BRI1 endosomal pool was observed in the *sgd1* mutant, suggesting that SGD1 may play an opposing role to PUB12/13 in regulating the stability of BRI1. Whether SGD1 collaborates with PUB12/13 or modifies the ubiquitination sites of BRI1 to regulate its endocytosis and degradation in foxtail millet remains unclear and will be the focus of our future study (lines 462-484).

Comment 12: *The Discussion could also make more of the use of SGD1 haplotypes in crop improvement instead of general comments about the Setaria system.*

Response 12: We appreciate your suggestion. The effect of *SGD1* haplotypes on crop improvement is the novelty of our research. This topic was further discussed (lines 510-518). For the *Setaria* system, although it has been proposed for years, very few reports that have really used it as an experimental system. In addition, we know maize and wheat are very important crops, but their genomes are complicated, which limits gene mining in these economically vital Poaceae crops. While in the *Setaria* system, forward gene cloning is as easy as that of *Arabidopsis*. As *S. italica* belongs to Poaceae family and is also a grain crop, it is a good choice to accelerate gene mining in maize and wheat. Therefore, we would like to declare this in discussion. Considering your suggestion, we deleted general comments about the *Setaria* model system and discussed aspects of crop improvement (lines 531-543).

Comment 13: *Methods. These are thoroughly described*

Response 13: Thank you.

Comment 14: *Figures. These are very well presented.*

Response 14: Thank you for the commendation on the quality of figures.

Comment 15: *Figure 1 shows grain number per panicle and 1000-grain weight as a key yield trait. In Figure 2 only grain size is used for the rice and wheat analyses. It is important to include grain number per panicle in Figure 2 to show the effect of SGD1 on overall yield.*

Response 15: This is a good suggestion. We agree that panicle-related traits should be included in the main result. Fortunately, we kept mature panicles of transgenic rice and wheat. In the revised manuscript, we added images of rice and wheat panicles and measured panicle weight, panicle length, and grain number per panicle. To our expect, these panicle-related traits showed significant declines in rice and wheat *SGD1* knockout plants, suggesting that *SGD1* also regulates panicle development in rice and wheat. Interestingly, we found that wheat *Tasgd1* triple mutant also showed shorter awn length than that of wildtype plants (revised Fig.3, Supplementary Fig. 5, and also in response Fig.1 as below). We added related content in revised manuscript (lines 168-170; lines 180-

Response Figure 1. SGD1 regulates panicle traits in rice and wheat. (A) Panicles of wild-type (WT) (KitaaKe), CR-Ossgd1-L1, and CR-Ossgd1-L2 rice plants. Bar = 2 cm. (B) Mature grains of WT (KitaaKe), CR-Ossgd1-L1, and CR-Ossgd1-L2 plants. Bar = 4 mm. (C). Panicle length and weight and grain number per panicle in rice plants (N = 5). Significant differences were determined using a two-sample *t*-test. ***P* < 0.001 vs. WT plants). (D) Panicles of WT (Fielder), CR- *Tasgd1a/1b/1d*-L1, and CR- *Tasgd1a/1b/1d*-L2 wheat plants. Bar = 4 mm. (e) Mature grains of WT (Fielder), CR- *Tasgd1a/1b/1d*-L1, and CR- *Tasgd1a/1b/1d*-L2 plants. Bar = 3 cm. (f). Panicle length and weight, grain number per panicle, and awn length in wheat plants (N = 5). Significant differences were determined using a two-sample *t*-test. ***P* < 0.001 vs. WT plants.

Comment 16: Figure 3. There needs to be an optical cross-section of FM4-64 and GFP signals to show co-localization.

Response 16: Thank you for the suggestion. We repeated subcellular location experiments and added a new figure showing that the SGD1-GFP signal colocalized with FM4-64 (revised Fig. 2g, also in response Fig. 2A as below). Colocalization was also assessed by measuring fluorescence intensity across the cell (revised Fig.2h and response Fig.2B).

Response Figure 2. Subcellular localization of SGD1. (A) SGD1-GFP colocalized with

the membrane marker FM4-64. Bar = 20 μ m. (B) Fluorescence intensity of SGD1-GFP and FM4-64 across the cell.

Comment 17: Figure 6 panel H should include a *bri1* ko control to show the specificity of the assays of BRI1 levels using SiBRI1 antibodies. Also, relative BRI1 levels could be quantified by scanning.

Response 17: That is indeed a very important suggestion which we have also considered before. We have been trying for two years to get a *bri1* ko plant in foxtail millet using CRISPR-Cas9 method, but we cannot get a homozygous mutant yet. We suspect that BRI1 KO is lethal in foxtail millet. Thus, we designed another experiment to assess the specificity of the SiBRI1 antibody. In our previous publication, we have already proved that anti-SiBRI1 antibody can recognize SiBRI1 in foxtail millet (Zhao et al., 2020). In the current study, to confirm the specificity of this antibody, we blasted the SiBRI1 amino acid sequence against foxtail millet protein databases and found that SiBRI1.L1 (Seita.9G296000), SiBRI1.L2 (Seita.2G165600), and SiBRI1.L3 (Seita.6G117300) were highly similar to SiBRI1 (Supplementary Figure 10, also in response Fig.3A). The coding sequences of SiBRI1, SiBRI1.L1, SiBRI1.L2, SiBRI1.L3 were cloned into the pCAMBIA1305-eGFP vector and transfected into foxtail millet protoplasts, respectively. After incubate for 10 hours, total protein was extracted and analyzed by western blotting. As shown in response Fig.3B, all these four proteins (SiBRI1-GFP, SiBRI1.L1-GFP, SiBRI1.L2-GFP, and SiBRI1.L3-GFP) were successfully detected using GFP antibody. Meanwhile, when we use SiBRI1 antibody, only SiBRI1-GFP can be recognized, other BRI1-like proteins cannot. This experiment clearly demonstrated the specificity of SiBRI1 antibody we used in our study (Supplementary Figure 10, and response Fig.3B as below). Given that we do not have *bri1* ko plant in foxtail millet, we hope this experiment can be an ideal alternative solution in verified specificity of SiBRI1 antibody. These results were described in lines 336-337.

As per your suggestion, we quantified relative BRI1 levels using ImageJ software, and added values in related figure (revised Fig. 6)

Response Figure 3. Phylogenetic analysis of BRI1-related LRR proteins and the

specificity of anti-SiBRI1 antibodies. (A) Phylogenetic analysis of *Setaria italica* BRI1 and its homologs. The red font represents SiBRI1, and the green font represents SiBRI1.L1, SiBRI1.L2, and SiBRI1.L3, which were highly similar to SiBRI1. (B) Specificity of anti-SiBRI1 antibodies. The coding sequences of SiBRI1, SiBRI.L1, SiBRI.L2, SiBRI.L3 were cloned into the pCAMBIA1305-eGFP vector and transfected into foxtail millet protoplasts. SiBRI1-GFP, SiBRI1.L1-GFP, SiBRI1.L2-GFP, and SiBRI1.L3-GFP were detected by anti-BRI1 and anti-GFP antibodies, respectively. Actin was used as a loading control.

Reference (for reviewer 1)

Chen, Q., Liu, R., Wu, Y., Wei, S., Wang, Q., Zheng, Y., Xia, R., Shang, X., Yu, F., Yang, X., Liu, L., Huang, X., Wang, Y., and Xie, Q. (2021). ERAD-related E2 and E3 enzymes modulate the drought response by regulating the stability of PIP2 aquaporins. *Plant Cell* 33, 2883-2898.

Jaillais Y, Vert G. Brassinosteroid signaling and BRI1 dynamics went underground. *Curr Opin Plant Biol.* 2016 Oct;33:92-100.

Martins S, Dohmann EM, Cayrel A, Johnson A, Fischer W, Pojer F, Satiat-Jeunemaître B, Jaillais Y, Chory J, Geldner N, Vert G. Internalization and vacuolar targeting of the brassinosteroid hormone receptor BRI1 are regulated by ubiquitination. *Nat Commun.* 2015 Jan 21;6:6151. doi: 10.1038/ncomms7151.

Tian X, He M, Mei E, Zhang B, Tang J, Xu M, Liu J, Li X, Wang Z, Tang W, Guan Q, Bu Q. WRKY53 integrates classic brassinosteroid signaling and the mitogen-activated protein kinase pathway to regulate rice architecture and seed size (2021). *Plant Cell.* 33(8):2753-2775.

Yang, J. et al. Brassinosteroids Modulate Meristem Fate and Differentiation of Unique Inflorescence Morphology in *Setaria viridis*. *Plant Cell* 30, 48-66 (2018).

Zhao M, Tang S, Zhang H, He M, Liu J, Zhi H, Sui Y, Liu X, Jia G, Zhao Z, Yan J, Zhang B, Zhou Y, Chu J, Wang X, Zhao B, Tang W, Li J, Wu C, Liu X, Diao X. DROOPY LEAF1 controls leaf architecture by orchestrating early brassinosteroid signaling (2020). *Proc Natl Acad Sci U S A.* 117(35):21766-21774.

Zhang, H., Zhou, J.F., Kan, Y., Shan, J.X., Ye, W.W., Dong, N.Q., Guo, T., Xiang, Y.H., Yang, Y.B., Li, Y.C., Zhao, H.Y., Yu, H.X., Lu, Z.Q., Guo, S.Q., Lei, J.J., Liao, B., Mu, X.R., Cao, Y.J., Yu, J.J., Lin, Y., and Lin, H.X. (2022). A genetic module at one locus in rice protects chloroplasts to enhance thermotolerance. *Science (New York, N.Y.)* 376, 1293-1300.

Zhang L, Liu Y, Zhu XF, Jung JH, Sun Q, Li TY, Chen LJ, Duan YX, Xuan YH. SYP22 and VAMP727 regulate BRI1 plasma membrane targeting to control plant growth in Arabidopsis. *New Phytol.* 2019 Aug; 223(3):1059-1065.

Zhou J, Liu D, Wang P, Ma X, Lin W, Chen S, Mishev K, Lu D, Kumar R, Vanhoutte I, Meng X, He P, Russinova E, Shan L. Regulation of Arabidopsis brassinosteroid receptor BRI1 endocytosis and degradation by plant U-box PUB12/PUB13-mediated ubiquitination. *Proc Natl Acad Sci U S A.* 2018 Feb 20;115(8): E1906-E1915.

Reviewer #2 (Remarks to the Author):

General comments

Overall, a strong study. It adds to an emerging story that the ubiquitin pathway impacts fundamental processes influencing agronomic traits. Not surprising result given the body of work in general that has emerged implicating the ub pathway is just about every process! Here the work is strong is demonstrating that SGD1 in Setaria is responsible for the phenotype, demonstrating that the related gene in other grasses works similarly. The significance of this effect is strengthened by creating mutants in wheat and rice, demonstrating its conservation. The effect of loss of SGD1 is supported by multiple mutants, and complementation with the coding region. Its biochemical function as an E3 ligase is confirmed.

Other strengths are identifying an interacting E2 and showing genetically its involvement in the same process. An impactful aspect is the allelic variation that can quantitatively influence the phenotype. Finally, the authors showed that SGD1 interacts with and ubiquitinates the brassinosteroid receptor, BRI1. The effect of this modification is not the canonical facilitation of degradation, but rather the protein is present at higher levels in presence of SGD1. This work will be of interest to both basic and applied researchers.

Despite many strengths, I do have several concerns. First, the authors seem overly concerned with identifying SGD orthologs (as opposed to related- homologs) in other species. The work in identifying functional orthologs in grasses is supported by their experimental evidence (mutational analyses through KOs in other species), but in dicots the identity of orthologous protein is much more tenuous, and I think should not be stressed. The authors try to extend the orthology comparisons to subcellular localization and, in part because their intracellular localization data are weak, the comparison seems forced and without sufficient support. Second, the subcellular localization data are not convincing, but they are not critical to the main impactful results, so could remove without affecting the conclusions. They do support the overall model that BRI1 abundance is affected in the ER or PM. The conclusion that SGD1 affects BRI1 stability could have stronger evidence. Third, there are multiple instances of incorrect or overinterpretation of the data. See below for more complete description/specifics of these concerns.

Response to general comments

Firstly, we would like to thank the reviewer for reviewing and appraising our study. The reviewer confirmed our efforts in revealing the SGD1-related ubiquitin pathway and creating multiple mutants in *Setaria*. This really encouraged us a lot, because performing these experiments in an orphan crop is more challenging than in prevailing models such as *Arabidopsis* and rice. Then, the reviewer also recognized our work on verifying SGD1's function in other grasses including rice and wheat. This is an important result that we highly value and put a lot of effort into. Moreover, in the revised manuscript, we added more figures and data on variations of panicle in *sgd1* rice and wheat mutants according to another reviewer's suggestion (revised Fig. 3). I think you may also feel interesting.

Thank you for the critical assessment and suggestions, which is really helpful in improving

the overall quality of the manuscript and increasing our understanding of the research field. We revised the manuscript thoroughly after carefully considering your comments and suggestions.

Specific comments

Comment 1: *Lines 82-102. This paragraph in the introduction describes BR signaling and the current knowledge of the intersection between ubiquitin pathway and BR pathway. In addition, it describes two BR-related mutants in Setaria. I think the order of topics within the paragraph is backwards. To aid the general reader, I suggest that, after the first sentence (revised as suggested below in 'Minor' section), the authors start with a description of BR signaling in general (using already existing sentences, starting at end of line 90). In other words, start the paragraph with the general BR response pathway, lines 90-102. I don't think the descriptions of the 2 Setaria BR mutants are relevant in the introduction and should be removed. The mutant dpy1 used in the work is mentioned in the appropriate section (but should have a citation) when utilized in relation to testing the role of BR signaling.*

Response 1: Thank you for these specific and thorough recommendations. We rewrote this paragraph following your suggestions (lines 71-87). The description of the two *Setaria* BR mutants was removed from the introduction. In addition, the *dpy1* mutant and phenotypes related to BR were mentioned and cited in the results section prior to the application of BR treatment (lines 258-262).

Comment 2: *Lines 100-102. What is the evidence that "there should be other E3 ligases that directly regulate either the protein stability, activity, or localization of the BRI1 receptor..."? I think the reader should be provided with some evidence to substantiate this claim that there should be other E3 that regulate BRI1 receptor function. This evidence could come from mutational studies, for example. For example, what is the regulation of BRI1 in a pub12/13 double mutant background? If it is still degraded, then this result indicates that additional activity also regulates BRI1 abundance. I think the rationale for the current study would be strengthened by providing some experimental support for the hypothesis that there should be other E3s regulating BRI1.*

Response 2: That is an interesting point. As previously reported in *Arabidopsis*, BRI1 ubiquitination in *pub12/13* mutants were significantly reduced compared to wild-type plants, resulting in the accumulation of BRI1 proteins. However, BRI1 internalization and degradation were still observed in *pub12/13* and the double mutant had little altered BR sensitivity (Zhou et al., 2018). These results indicated that, besides PUB12/13, there may be other E3 ligase(s) that directly regulate either the protein stability, activity, or localization of the BRI1 receptor through ubiquitination modification, which remains to be explored. We have added this content in the text (lines 78-85).

Comment 3: *Results, lines 167-170. The authors write, "Phylogenetic analysis suggested that SGD1 is an ortholog of Arabidopsis PPRT1 and rice TT3.1 gene (Supplementary Figure S3), which were reported to be involved in salt and high temperature stress responses." Supp. Fig S3 does not provide evidence of orthology to these specific proteins.*

This is because only proteins with shared domains are included, so a greater identity to these proteins over other E3 is not shown. There are 3 Arabidopsis proteins in the figure, which is PPRT1? Next, which of the 3 Arabidopsis proteins is most similar to SGD1? They all branch off together, so I don't see how you can say one of these 3 is the ortholog. Could the authors clarify?

There is only one rice protein in the analysis, so we can't evaluate orthology. A statement that TT3 is the only protein with these 2 domains and has the highest identity to SGD1 over all other rice proteins are evidence to support orthology. Because of the uncertainty, making direct comparisons of SGD1 to PPRT1 localization and function seems problematic.

Response 3: Thank you for the professional suggestion. We agree that the original phylogenetic analysis and the identification of *SGD1* orthologs are weak. To address this concern, we made a genome-wide analysis of the RING-type E3 ubiquitin ligase gene family. Previous studies identified 469 RING domain-containing proteins in *Arabidopsis* (Stone et al., 2005), and 476 RING-type proteins in rice (Lim et al., 2010; Wang et al., 2022). Using a previously described method (Lim et al., 2010), we identified 495 RING-type proteins in foxtail millet (Supplementary Table 5). These proteins can be divided into eight subfamilies according to the RING domain type (Stone et al., 2005). Among them, *SGD1* belongs to a C3HC4 RING-type ubiquitin ligase subfamily.

To comprehensively figure out the relationship between *SGD1* and E3 homologs, annotated protein databases of *Arabidopsis*, *Setaria italica*, *Oryza sativa*, *Triticum aestivum*, and *Zea mays* were obtained from Phytozome (<http://www.phytozome.net>). The profile hidden Markov model of the C3HC4 RING-finger domain was obtained from the Pfam database (PF13920). The HMMER program (<http://hmmer.org/>) was used to search against reference protein databases with default parameters. HMMER-selected proteins were then scanned for the presence of the C3HC4 Zinc finger RING-type domain (IPR018957) by InterProScan (<https://www.ebi.ac.uk/interpro/about/interproscan/>). Finally, we identified a total number of 240 C3HC4 RING-type E3s in selected plant species, with 38 in *Arabidopsis*, 39 in *S. italica*, 40 in *O. sativa*, 53 in *Z. mays*, and 70 in *T. aestivum*. Blast identity values between *SGD1* and other E3s were listed in (Supplementary Table 6). To examine the phylogenetic relationship among these E3s, a maximum likelihood phylogenetic tree was constructed using full-length amino acid sequences. The phylogenetic analysis clearly showed that TT3.1 was the only rice E3 which was divided into the same branch of *SGD1* (Supplementary Figure 4, also in response Figure 4 as below). Combine with the BLAST identity shown in Supplementary Table 6, we confirmed that rice TT3.1 is the ortholog of *SGD1*.

In *Arabidopsis*, the result is more complicated. All these three genes including *AT1G68820* (*PPRT1*), *AT1G18470* (*PPRT3*), and *AT1G73950*, were divided into the same branch of *SGD1* (Supplementary Figure 4, response Figure 4). Their BLAST identities were also very close (Supplementary Table 6). We suspected that these genes underwent duplication. A search against the Plant Duplicate Gene Database (Lee et al., 2017) confirmed our hypothesis. As shown in response Figure 5, the expansion of *SGD1*'s orthologs in *Arabidopsis* has occurred mainly through dispersed duplication and whole-genome

duplication. We also revised the statement to clarify the confusion. ‘Phylogenetic analysis suggests that SGD1 is orthologous to TT3.1 (Supplementary Fig. 4), which was reported to be a positive regulator in promoting rice thermo-tolerance. In addition, three orthologs of SGD1 were identified in *Arabidopsis* based on the phylogenetic tree (Supplementary Fig. 4) and BLAST identity (Supplementary Table 6), among which PPRT1 (AT1G68820) was associated with salt stress response.’ (Lines 163-167).

Response Figure 4. Phylogenetic analysis of SGD1 and other E3s in *Arabidopsis thaliana*, *Setaria italica*, *Oryza sativa*, *Zea mays*, and *Triticum aestivum*. Full-length amino acid sequences of C3HC4 E3s (38 in *Arabidopsis*, 39 in *S. italica*, 40 in *O. sativa*, 53 in *Z. mays*, and 70 in *T. aestivum*) were identified. Sequences were aligned using MUSCLE in MEGA X software and were used to construct an unrooted phylogenetic tree based on a maximum likelihood method after bootstrap analysis for 1000 replicates. The red branch represents SGD1 and its orthologs.

AT1G18470.1

You can search duplicate gene pairs by entering gene id, e.g. Carubv10012323m, Bra022826, BnaC07g23860D, AALP_AA1G000800

Duplicate1	Location1	Duplicate2	Location2	Evalue	Type
AT1G18470.1	Ath-Chr1:6356131	AT1G68820.1	Ath-Chr1:25865660	0.0	dispersed
AT1G18470.1	Ath-Chr1:6356131	AT1G73950.1	Ath-Chr1:27799830	0.0	wgd

Response Figure 5. Orthologs of SGD1 duplicated in Arabidopsis. Gene duplication events of Arabidopsis genes AT1G68820 (PPRT1), AT1G18470 (PPRT3), and AT1G73950 were searched in the Plant Duplicate Gene Database (<http://pdgd.njau.edu.cn:8080/>).

Comment 4: Lines 210-212. Related to protein relationships, the authors did not provide us with information in support of their conclusion that *Seita.9G428900* is the ortholog of *UBC32*. From sequence comparisons, it is most similar *UBC32*

Response 4: This is a similar problem to comment 3. We thus make a genome-wide analysis of all ubiquitin-conjugating enzymes (E2s) in both Arabidopsis and foxtail millet. E2 enzymes featured with its ubiquitin-conjugating (UBC) domain (PF00179; IPR000608; Jentsch et al., 1990). Combining HMMER and BLAST to search against the Arabidopsis and foxtail millet protein databases (downloaded from Phytozome), we identified 37 E2s in Arabidopsis and 41 E2s in foxtail millet. The number of E2s we identified in Arabidopsis is consistent with a previous report (Kraft et al., 2005), indicating our procedure in identifying gene family members is robust. Furthermore, phylogenetic analysis of the complete amino acid sequences of these 78 E2s showed that *Seita.9G428900* was orthologous to Arabidopsis *UBC32* (response Figure 6 as below). Related content was also included in the revised manuscript (line 189-191, Supplementary Figure 6).

Response Figure 6. Phylogenetic analysis of Arabidopsis and foxtail millet E2

enzymes. The complete amino acid sequences of 78 E2 enzymes were aligned using MUSCLE and analyzed phylogenetically using MEGA X software. An unrooted phylogenetic tree was constructed using the neighbor-joining method after bootstrap analysis for 1000 replicates. The branch containing UBC32 is highlighted in red.

Comment 5: *The results from the intracellular localization experiments are weak (Fig 3C,D). The lack of a merged color in 3C suggests that the fusion protein is NOT localized to the PM (or the FM signal is overwhelming the GFP signal). Cells with more evident cytoplasm should be used to assign localization more definitively.*

Response 5: We agree that the original Figure 3C on SGD1-GFP colocalization with FM4-64 is weak. Previous research on SGD1's rice ortholog TT3.1, suggested that the protein was located in plasma membrane (PM) (Zhang et al., 2022), but we found that SGD1 was not only located in PM. Therefore, we used that figure to show some signals that not merged with FM4-64. Now, we realized that it does make confusion to readers. Considering your suggestion, we deleted the Fig. 3C from the main text to avoid confusion. In addition, combined with the suggestion raised by the other reviewer, we repeated the subcellular location experiment, and got a better result which showed that the SGD1-GFP signal merged well with FM4-64 (revised Fig.2g). This result is similar to that of in rice (Zhang et al., 2022). We also confirmed that the ER location of SGD1 is correct through multiple experiments, thus we kept the result on SGD1-GFP co-location with the ER marker (HDEL-mCherry) in the main text (revised Fig.2i). In addition, colocalization was also assessed by measuring fluorescence intensity across the cell (revised Fig.2h and 2j). The related description was also revised (lines 140-149).

Comment 6: Lines 249-264 and Figure 4. The authors generate a double *ubc32 sgd1* mutant and conclude that the two genes act synergistically to regulate grain yield. I don't see this synergism. It is difficult to evaluate from the text because we are not given values for the single mutants (please include the numbers for the single mutants), but looking at the graphs, the double mutant is more severe than each single, but the effect appears additive, not synergistic. Could the authors provide more support for their synergistic hypothesis? It would also be of value to provide a model. If they both work together and are required together for ubiquitination of BRI1, then I would expect the double mutant to look like each of the single mutants, not to have a greater effect.

Response 6: Thank you for pointing out this issue. We misused the word "synergistically" to describe the relationship between SGD1 and SiUBC32. According to your suggestion, we added a supplementary table (Supplementary Table 8) to evaluate the main agronomic traits of WT, double mutant, and each single mutant. The raw data containing values of all biological replications of each trait are included in the source data file (sheet- fig. 4k). We also briefly summarized these changes in Response Table 1 as below. From these data, we can clearly see that the double mutant is more severe than each single mutant in all investigated traits. We further compared the reductions observed in two single mutants with that in the double mutant. We found that the reductions of plant height, flag leaf length, and panicle length in the double mutant are much greater than each single mutant, suggesting these two genes act additively in regulating these three traits. However, for

grain yield-related traits, including 1000-grain weight, grain length, and grain width, the double mutant is slightly more severe than each single mutant. Take 1000-grain weight for example, *sgd1* showed a 40.0% decline, *Siubc32* exhibited a 40.5% reduction, and the double mutant showed a 43.2% reduction, which is only slightly greater than each single mutant. This result suggests that SGD1 and SiUBC32 have partially overlapping functions in regulating grain yield. A similar result was published in describing the genetic relationship between Large1 and GSK2 (Lyu et al., 2020). At the biochemical level, we also found that SGD1 and SiUBC32 are partners in an E2-E3 complex, which further modified their target proteins such as SiBR11. However, we should recognize that SGD1 and SiUBC32 are not simply acting in the same linear pathway or as a single complex. There are other E3s that can interact with UBC32 (Chen et al., 2021). There are also other target proteins that can be regulated by SGD1 besides BR11 (Zhang et al., 2022). Combining our result with previous reports, we conclude that SGD1 acts, at least in part, in a common genetic pathway with SiUBC32 to control grain-yield related traits. Related content was added in the revised manuscript (lines 226-256).

Response Table 1. Comparison of agronomic trait change between wild-type Ci846, *sgd1*, *Siubc32*, and *sgd1/Siubc32* double mutants.

Agronomic trait	Percentage (sgd1 to WT)	Percentage (Siubc32 to WT)	Percentage (sgd1/Siubc32 to WT)
Plant height	-24.92%	-31.27%	-51.29%
Flag leaf length	-25.93%	-31.43%	-52.75%
Panicle length	-34.00%	-37.24%	-66.37%
1000-grain weight	-40.00%	-40.50%	-43.23%
Grain length	-11.36%	-13.72%	-17.50%
Grain width	-22.29%	-22.40%	-24.08%

Values are means of five independent biological replications for plant height, flag leaf length, panicle length, and 1000-grain weight. Values are means of >100 independent biological replications for grain length and width. The percentage represents the reduction of each trait compared to the wild-type plant.

Comment 7: Lines 282-285. The following statement seems incorrect in concluding that *sgd1* is insensitive to BL. The authors write, “A clear inhibition effect was observed at 0.01 μ M and 0.1 μ M eBL treatment in WT seedlings, whereas the corresponding effects were not seen in the *sgd1* mutants (Figure 5F and G). These results indicated that *sgd1* is insensitive to BR.” In the panel (G), application of eBL does result in shorter roots in the *sgd1* mutant, but the effect is not as strong as in wild type roots. This result indicates that the roots are not insensitive, but less sensitive (hyposensitive) to exogenous BL. Authors please correct. (the overall conclusion that *sgd1* has reduced response to BL – lines 308-310 – is well supported by the data.

Response 7: Thank you for pointing out the unintentional mistake. We agree that the *sgd1* mutant is less sensitive to exogenous BL, not insensitive. The text was revised accordingly (lines 274, 344, 449-450).

Comment 8: Figure 6F. From the data in this panel, the authors conclude that “SiBR11 ubiquitination and protein stability were reduced in *sgd1* compared to WT plants”

(statement in 6F legend). The figure only shows reduced ubiquitination, not stability. The experiment immunoprecipitates ubiquitinated proteins in total, then visualizes in that IP pool tagged- BRI1. They can only assess the level of ubiquitinated forms.

Response 8: Yes, we totally agree with you that the Figure 6F can only show that SiBRI1 ubiquitination was reduced in the mutant. We have amended related content in Fig. 6f legend (lines 933-934). In addition, to determine SiBRI1 protein stability, we performed new experiments as you suggested in comment 10. These results indicated that SiBRI1 stability was lower in *sgd1* mutants than in WT plants. Please refer to response 10 for further details.

Comment 9: Discussion, lines 443-449. The authors give a list of reasons in support of their conclusion that “interaction and ubiquitination process led to enhancement of BRI1 accumulation and BR signaling rather than degradation of BRI1 for these following reasons.” The first two reasons listed are not in support of this conclusion. They support the hypothesis that SGD1 ubiquitinates BRI1 in vivo, not that SGD1 leads to enhancement of BRI1 accumulation and BR signaling. Evidence that SGD1 enhances BRI1 accumulation and BR signaling are the observed hypo-sensitivity of *sgd1* mutants to BR, that levels of BRI1 appear to be higher when SGD1 is over-expressed, that ubiquitination is lower in *sgd1* mutants and BZR1 phosphorylation status is affected (these arguments are included further down in the discussion). These first statements should be removed or included in a prior conclusion that BRI1 is an in vivo substrate of SGD1.

Response 9: Thank you for pointing out this inconsistency. We removed the first two reasons that are not in support of related conclusion and reorganized the text (lines 443-455).

Comment 10: The main claim mechanistically is that SGD1 ubiquitinates and stabilizes BRI1. This is supported by some data that are not as conclusive as one would like. While there are several experiments consistent with this interpretation, there is no direct measurement of BRI1 protein stability. Given that they have Ab to the *Setaria* BRI1, the authors could immunoblot for BRI1 in WT and in multiple *sgd1* mutant lines to compare BRI1 abundance. Important to demonstrate the effect in all the mutant lines. Second, the authors should show that co-expression of the catalytically inactive form of SGD1 does not affect BRI1 accumulation (add to experiment shown in Fig 6G). The authors should perform a cycloheximide experiment- to measure stability by monitoring protein abundance over time in the presence of cycloheximide. It should be possible with plant roots (for example since immunoblotting extracts yields signal-see Fig 6H). Or the same protocol could be done using protoplasts after transfection --- incubate for expression – then add cycloheximide and monitor BRI1 abundance over time +/- coexpression with SGD1. (Figure 6G experiment with a cycloheximide time course).

Response 10: We greatly appreciate these suggestions. We designed three experiments according to your directions, and the results increased the reliability of the conclusions. First, we compared SiBRI1 abundance in four different *sgd1* mutant lines (*CR-sgd1-L1/L2* under a Ci846 background, *sgd1-1/2* under a Yugu1 background). Total protein was isolated from 14-day-old WT plants and four mutant lines. Endogenous SiBRI1 was detected by immunoblotting with an anti-SiBRI1 antibody. As shown in Fig. 6i and

Supplementary Fig. 11 (also summarized in Response Figure 7A as below), SiBRI1 abundance was lower in four different *sgd1* mutants compared to corresponding WT plants, indicating loss of function of SGD1 led to reduced SiBRI1 stability. Second, we co-expressed both the catalytically active and inactive form of SGD1 with SiBRI1 in foxtail millet protoplasts. The result indicated that the catalytically active form of SGD1 increased the levels of SiBRI1 protein, while the inactive form of SGD1 does not affect SiBRI1 accumulation (Fig. 6g or Response Figure 7B). This result indicates SiBRI1 protein stability depends on catalytic activity of SGD1. Third, we performed cycloheximide (CHX) treatments. SiBRI1 protein stability in WT (Ci846), *CR-sgd1* mutants, and *OE-SGD1* plants was assessed by monitoring protein abundance over a 6-hour period after CHX treatment. The relative intensities of the SiBRI1 protein level were evaluated based on three independent biological replicates. CHX time-dependently decreased SiBRI1 levels in WT and mutant plants, and the decrease was more pronounced in the mutant lines. It is worth noting that in the presence of CHX, the level of SiBRI1 remained stable in SGD1-overexpressing lines, with only a slight decreasing trend observed (Fig. 6k-p, Supplementary Fig. 12, and also in Response Figure 7C as below). To sum up, these results strongly support the conclusion that SGD1 stabilizes BRI1 (lines 345-358).

Response Figure 7. Analysis of SiBRI1 stability. (A) SiBRI1 stability was reduced in *sgd1* mutant lines. SiBRI1 was detected in 14-day-old Ci846, *CR-sgd1*-L1, *CR-sgd1*-L2, Yugu1, *sgd1*-1, and *sgd1*-2 seedling leaves immunoblotted with an anti-SiBRI1 antibody. The relative abundance of SiBRI1 is shown above the blots. (B) SGD1 enhances SiBRI1 protein stability in *S. italica* protoplasts. SiBRI1-HA, SGD1-FLAG, and mSGD1-FLAG were detected by immunoblotting with anti-HA and anti-FLAG antibodies, respectively. GFP-Myc was used as an internal transfection control and detected by immunoblotting with anti-Myc antibody. The relative intensity of SiBRI1-HA protein abundance was shown above the blot bands. (C) SiBRI1 abundance in 14-day-old Ci846, *sgd1*, and *OE-SGD1* seedling leaves treated with 100 μ M cycloheximide for 0, 3, and 6 h. SiBRI1 was measured by immunoblotting with an anti-SiBRI1 antibody. Actin was used as a loading control. Three biological replicates were used for each treatment (Supplementary Figure 12).

Minor. Typos. use of wrong word

1. Line 58-59, I think there is a typo- "is one of the examples that extent knowledge" should

be “is one of the examples that extends knowledge”?

Response: Yes, we correct it (line 57). All the typos were fixed according to your suggestions. In addition, the manuscript was carefully revised by two native English speakers. The language editing certificate was uploaded as a supplemental file.

2. Line 74-75 modify, “that work redundantly in controlled seed size through regulating cell...” to “that work redundantly in controlling seed size through regulating cell...”

Response: Related content was revised (line 66).

3. Lines 79-81, not sure of meaning of this sentence: “Beyond expectation, as E3’s working partner, few E2 was identified to be involved in grain yields regulation, let alone any E2/E3 pairs in plants.” I think that this statement, if I think I understand its meaning, should just be deleted. E2/E3 pairings have been reported in plants, and what does “few E2” mean? I don’t think this statement provides a strong argument in support of the significance of the work reported here and should just be removed.

Response: Yes. This sentence was deleted as requested.

4. Line 83, “arrange” should be “range”. The sentence containing “arrange” is awkward even with correction. How about “Brassinosteroid (BR) is a phytohormone that broadly regulates plant growth and development, as well as stress responses.” All phytohormones are important, so I don’t think that term should be included.

Response: This statement was revised (lines 71-72).

5. line 86, change “homolog” to “homologous”

Response: This sentence was removed according to another reviewer’s suggestion.

6. lines 106-107, revise “Homologous comparison suggested” to “Homology comparisons suggest”

Response: The sentence was rephrased to “phylogenetic analysis suggests” (line 91-92).

7. Line 114, modify “genetical” to “genetic”

Response: This error was corrected (line 98)

8. Line 115, modify “support the” to “support that the”

Response: Yes. It was fixed (line 98).

9. Line 116, modify “grain yields” to “grain yield”

Response: We modified it (line 99).

10. Line 116-117, modify “high-yields breeding” to “high-yield breeding”

Response: Thank you. We fixed it (lines 100).

11. Figure S3. Could the gene names be rotated so they are as right-side-up as possible? Hard to read the figure when many are upside down.

Response: Yes, we reproduced this figure according to your suggestion (revised Supplementary Fig. 4).

12. Figure S3. What is the species represented by EI?

Response: EI represents a plant species named Humans used beets (*Beta vulgaris* spp. *vulgaris* L.). Gene ID of this species in Phytozome starts with 'EI'. In the revised manuscript, we deleted this species in related content (Supplementary Fig. 4).

13. Figure S3. Could the authors include in the legend the abbreviations used for each species- in parenthesis after the name- for example, Zea mays, Zm...

Response: Yes, we included legend for all abbreviations in Supplementary Fig. 4.

14. Line 168, could the AGI number for PPRT1 be given in the text? Because Fig S3 only shows AGI numbers for the Arabidopsis proteins so the reader can't know which protein is PPRT1.

Response: Sorry that we did not make it clear. In the revised manuscript, the phylogenetic tree was reconstructed according to your suggestion in specific comment 3. The gene name and AGI number were shown in both figure and main text (line 167).

15. Line 170, "CRIPSER" should be "CRISPR"

Response: Yes, the typo was corrected (line 168).

16. Line 171, "ossgd1" should be "Ossgd1". There is more than one crispr line, so the text should reflect that information-such as, "two CRISPR-edited rice Ossgd1 lines both showed small grains (Figure 2G and H)."

Response: Thank you, we replaced the sentence accordingly (lines 168-169).

17. Line 172 "conservative" should be "conserved"

Response: We corrected it (line 171).

18. Line 173, "firstly isolated two SGD1 homologs" change to "first identified two SGD1 homologs"

Response: Thank you. We change the word (line 171).

19. Line 174, Not sure what is meant by "For its higher DNA similarity," Was this one chosen because its protein (not DNA) sequence had highest identity to SGD1? Please clarify.

Response: Yes. We did select the gene for its highest identity to SGD1 in protein sequence. Related content was corrected (lines 172-173).

20. Line 191, "tissues" should be replaced with "organs". The parts of the plant tested are different organs.

Response: Thank you for the correction. We change "tissues" to "organs" throughout the manuscript.

21. Line 197, "fusing" should be changed to "fusion"

Response: We amended it (line 139).

22. Line 202, I don't see yellow arrows in Fig. 3C. Maybe in reducing their size, the color differential between white and yellow is minimized? I suggest different shapes instead of different colors, one using an arrowhead only, another using an arrow.

Response: We deleted Fig. 3C according to your general comment.

23. Line 211. Modify "which is a kind of ubiquitin-conjugating enzyme" to "which is a ubiquitin-conjugating enzyme"

Response: We modified it (line 190).

24. Line 223, Modify "SiiUBC32" to "SiUBC32"

Response: Thank you. We fixed the typo (line 201).

25. For many lines throughout the text, the authors should capitalize the S of "Siubc32" and in all other mutant alleles because the Si of the name refers to the plant species and is not lowercase even in mutant allele names.

Response: Thank you for pointing out this. We corrected all these similar mistakes in describing the mutant name throughout the manuscript.

26. line 265, heading title, suggest modifying "Loss of function in SGD1 is involved in BR signaling" to "SGD1 is involved in BR signaling". The loss-of-function mutant in SGD1 allows you to discover that SGD1 has a role in BR signaling.

Response: Yes. We modified related content (line 257).

27. Line 273, Not sure what is meant by "isolated" ? Maybe could say, a naturally occurring active brassinolid? Just a suggestion, could the authors clarify?

Response: Yes, we want to tell 24-epi brassinolide is an active brassinolide isolated from plants. Thank you for proving us better words. We amended it accordingly (line 264).

28. Line 276, "while the Leaf angle" should be "while the leaf angle"

Response: Yes, we corrected it (line 267).

29. Line 287, "-taget" to "-target"

Response: We fixed typo (line 276).

30. Line 288, "Chang" to "Change"

Response: Thank you. We corrected it (line 278).

31. Line 295, "tuned" to "turned"

Response: We change it (line 284).

32. Lines 301-302. I think the authors mean the SGD1 plays a role in the negative feedback of BL biosynthesis, not the loss-of-function version, because the mutant has higher levels of mRNA for BL biosynthetic genes.

Response: Thank you, we deleted this sentence to avoid confusion.

33. Line 357, modify “BR-insensitive phenotypes” to “BR-hypo-sensitive phenotypes”

Response: Thank you. We changed it accordingly (line 344).

Reference (for reviewer 2)

Wang R, You X, Zhang C, et al. An ORFeome of rice E3 ubiquitin ligases for global analysis of the ubiquitination interactome. *Genome Biology*. 2022 Jul;23(1):154.

Stone SL, Hauksdóttir H, Troy A, et al. Functional analysis of the RING-type ubiquitin ligase family of Arabidopsis. *Plant Physiol*. 2005 Jan;137(1):13-30.

Lim SD, Yim WC, Moon JC, Kim DS, Lee BM, Jang CS. A gene family encoding RING finger proteins in rice: their expansion, expression diversity, and co-expressed genes. *Plant Mol Biol*. 2010 Mar;72(4-5):369-80.

Lee TH, Kim J, Robertson JS, Paterson AH. Plant Genome Duplication Database. *Methods Mol Biol*. 2017; 1533:267-277.

Lyu J, Wang D, Duan P, Liu Y, Huang K, Zeng D, Zhang L, Dong G, Li Y, Xu R, Zhang B, Huang X, Li N, Wang Y, Qian Q, Li Y. Control of Grain Size and Weight by the GSK2-LARGE1/OML4 Pathway in Rice. *Plant Cell*. 2020 Jun;32(6):1905-1918.

Jentsch S, Seufert W, Sommer T, Reins HA. Ubiquitin-conjugating enzymes: novel regulators of eukaryotic cells. *Trends in Biochemical Sciences*. 1990 May;15(5):195-198.

Kraft E, Stone SL, Ma L, Su N, Gao Y, Lau OS, Deng XW, Callis J. Genome analysis and functional characterization of the E2 and RING-type E3 ligase ubiquitination enzymes of Arabidopsis. *Plant Physiol*. 2005 Dec;139(4):1597-611.

Zhang H, Zhou JF, Kan Y, Shan JX, Ye WW, Dong NQ, Guo T, Xiang YH, Yang YB, Li YC, Zhao HY, Yu HX, Lu ZQ, Guo SQ, Lei JJ, Liao B, Mu XR, Cao YJ, Yu JJ, Lin Y, Lin HX. A genetic module at one locus in rice protects chloroplasts to enhance thermotolerance. *Science*. 2022 Jun 17;376(6599):1293-1300.

Zhou J, Liu D, Wang P, Ma X, Lin W, Chen S, Mishev K, Lu D, Kumar R, Vanhoutte I, Meng X, He P, Russinova E, Shan L. Regulation of Arabidopsis brassinosteroid receptor BRI1 endocytosis and degradation by plant U-box PUB12/PUB13-mediated ubiquitination. *Proc Natl Acad Sci U S A*. 2018 Feb 20;115(8):E1906-E1915.

Reviewer #3 (Remarks to the Author):

General comments

*The authors present an extremely thorough phenotypic and genetic analysis of an E3 ligase that influences grain yield in *Setaria italica* including mutant analysis, identifying interacting partners and investigating the role of brassinosteroids on the phenotype. They demonstrated that SGD1 orthologs in wheat and rice influence grain yield and that a maize ortholog can rescue the *Setaria* mutant. Thus, SGD1 is of broad importance in grain yield. They also conducted a population analysis that indicated the SGD1 locus was under selection during domestication and identified 3 haplotypes associated with domestication, including one that is dominant in elite cultivars. This haplotype is also associated with the greatest increase in grain yield. Overall, the experiments conducted were very thorough and supported the conclusions. While the figures are rather complex, they present the data in a comprehensible fashion and show numerous lines of evidence to support their conclusions. I think the extremely broad characterization spanning from protein and molecular characterization to other grasses and population analysis combined with the apparent importance of the gene in grain yield and domestication will make this work of great interest to readers. I have the following suggestions to improve the manuscript.*

Response to general comments

Thank you very much for acknowledging our efforts in identification and characterization SGD1 in *Setaria italica* and other important crops. Your positive comments not only help us in improving the current manuscript, but also encourage us to carry on functional genomic research in *Setaria*. We have carefully considered your suggestions and revised our manuscript point-by-point. Here, we list below our responses to the comments/suggestions.

Specific comments

Comment 1: *The manuscript needs to be carefully edited for grammar and language because there are many awkward sentences. I note a few below, but my list is not comprehensive.*

Response 1: Thank you for the appraisal. We fixed all language problems according to your and other two reviewers' suggestions. Moreover, the manuscript was thoroughly revised by two native English speakers. The language editing certificate was uploaded as a supplementary file. We hope the revised version meets the journal's language standards.

Comment 2: *The authors write that SGD1 and SiUBC32 are primary partners in an E2-E3 complex. However, their double mutant analysis showed that they act additively. This indicates that they are not simply acting in the same linear pathway or as a single complex. The authors should discuss this in the results and discussion (line 435) and perhaps postulate possible reasons for this interesting genetic observation and the fact that they clearly interact based on other evidence. I also noticed that the SGD1 band is much weaker than the SiUBC32 band in their pulldown experiments. That may be something they could postulate about.*

Response 2: We highly appreciate this comment. Your idea on ‘SGD1 and SiUBC32 are not simply acting in the same linear pathway or as a single complex’ is correct. In revised manuscript, we mainly proposed three points to discuss this interesting genetic observation. Firstly, we added a table (Supplementary Table 8) to show phenotype values for the WT, single mutants, and double mutants. The raw data containing values of all biological replications of each trait are included in the source data file (sheet- fig. 4k). We also summarized these changes in Response Table 1 as below. From these data, we can clearly see that the double mutant is more severe than each single mutant in all investigated traits. We further compared the reductions observed in two single mutants with that in the double mutant. We found that the reductions of plant height, flag leaf length, and panicle length in the double mutant are much greater than each single mutant, suggesting these two genes acts additively in regulating these three traits. However, for grain yield-related traits, including 1000-grain weight, grain length, and grain width, the double mutant is slightly more severe than each single mutant. Take 1000-grain weight for example, *sgd1* showed a 40.0% decline, *Siubc32* exhibited a 40.5% reduction, and the double mutant showed a 43.2% reduction, which is only slightly greater than each single mutant. This result suggests that SGD1 and SiUBC32 have partial overlapping functions in regulating grain yield. Similar result was published in describing the genetic relationship between Large1 and GSK2 (Lyu et al., 2020).

Response Table 1. Comparison of agronomic trait change between wild-type Ci846, *sgd1*, *Siubc32*, and *sgd1/Siubc32* double mutants.

Agronomic trait	Percentage (sgd1 to WT)	Percentage (Siubc32 to WT)	Percentage (sgd1/Siubc32 to WT)
Plant height	-24.92%	-31.27%	-51.29%
Flag leaf length	-25.93%	-31.43%	-52.75%
Panicle length	-34.00%	-37.24%	-66.37%
1000-grain weight	-40.00%	-40.50%	-43.23%
Grain length	-11.36%	-13.72%	-17.50%
Grain width	-22.29%	-22.40%	-24.08%

Values are means of five independent biological replications for plant height, flag leaf length, panicle length and 1000-grain weight. Values are means of >100 independent biological replications for grain length and width. The percentage represents the reduction of each trait compared to the wild-type plant.

Secondly, our biochemical experiments proved that SGD1 and SiUBC32 interacted with each other in vivo and in vitro. However, we should admit that SGD1 and SiUBC32 are not simply acting in the same linear pathway or as a single complex. One of the clues we noticed is that the SGD1 band is much weaker than the SiUBC32 band in their pulldown experiments. Previous report also supports our postulation (Chen et al., 2021). UBC32 can interacted and ubiquitinated another RING-type E3 ligase Rma1 in Arabidopsis, which finally regulated drought tolerance in Arabidopsis and rice. Thirdly, SiUBC32 is also not the only upstream E2 of SGD1, which could be supported by our Y2H screening result (Supplementary Table 7). Moreover, UBC32 and SGD1 may have their own target proteins besides BRI1. According to previous publications, Arabidopsis UBC32-Rma1 complex can ubiquitinate a phosphorylated form of PIP2 aquaporin and promotes its degradation (Chen

et al., 2021). OsSGD1 (TT3.1) interacts and ubiquitinates TT3.2 for degradation, which help rice to maintain high-yield under heat stress (Zhang et al., 2021). Combine our result with previous reports (Chen et al., 2021; Zhang et al., 2022), we conclude that SGD1 acts, at least in part, in a common genetic pathway with SiUBC32 to control grain-yield related traits. Related content was added in revised manuscript (lines 236-256).

Comment 3: *The interpretation of the results of the overexpression of SGD1 in cultivar in Ci846 is not straightforward because the gene inserted is a different haplotype than the native SGD1 gene in Ci846. Since the haplotype inserted is associated with higher yield than the native Ci846 haplotype, it cannot be concluded that the observed increase in yield is due to overexpression rather than to the haplotype. The authors should acknowledge this in the discussion and abstract.*

Response 3: Thank you for the thoughtful suggestion. We agree with this it. The SGD1 allele overexpressed in Ci846 is the elite haplotype associated with a higher yield, not the native haplotype, preventing assessing whether the observed increase in yield is due to SGD1 overexpression. To solve this problem, we need more experiments in the future. On one hand, we can overexpress SGD1 H1 and H3 haplotype in an *sgd1* mutant background, respectively, and compare the effects of these haplotypes on grain yield. On the other hand, we can construct nearly isogenic lines (NILs) that carrying the high-yield allele SGD1-H1 in the genetic background of the low-yield allele SGD1-H3 (e.g., Ci846) through repeated backcrossing and then comparing grain yield traits between NIL-SGD1^{H1} and NIL- SGD1^{H3}. Combine these results, we will get the contribution ratio of overexpression and haplotype to grain yield. But it will take years to finish these experiments. However, we can get some clues from the research of rice SGD1 (TT3.1, Zhang et al., 2022) on thermotolerance. In rice, haplotypes associated with resistance and susceptibility to heat stress (TT3.1^{CG14} and TT3.1^{WYJ}, respectively) were overexpressed in the WYJ variety (Zhang et al., 2022). The lines overexpressing TT3.1^{CG14} or TT3.1^{WYJ} were more resistant to heat stress than the wild-type line. Moreover, at similar gene expression levels, the line overexpressing the elite haplotype (TT3.1^{CG14}) was more resistant to heat stress than the line overexpressing TT3.1^{WYJ}, which indicates that both overexpression and elite haplotype contribute to thermotolerant, they worked additively in heat stress tolerance. We acknowledge this issue in both the abstract (line 36) and the discussion section (lines 513-517).

Comment 4: *The authors make numerous mentions of the Setaria model system. This may be confusing to many readers who are familiar with Setaria viridis as a model plant and Setaria italica as a grain crop. It would be clearer if the authors include an introductory sentence indicating that they are using foxtail millet, Setaria italica, as a model system for grains in general.*

Response 4: Thank you for your concern and support on Setaria model. The Setaria model system was proposed at the 1st International Setaria Genetics Conference (Beijing, China, 2014), where all experts agreed that this model system contained *S. viridis* and *S. italica*. These species are very similar and can cross with each other easily. Thus, a few experts believe that *S. viridis* and *S. italica* are genetically one species; for this reason, we did not distinguish these species in our study. Considering your suggestion, we changed 'Setaria

model system' into '*Setaria italica*' in its first occurrence (line 26), and included an introductory sentence in main text (lines 53-54).

Comment 5: *Line 73: They should add references for this sentence "A few ubiquitin-related proteins involved in grain yield control were identified."*

Response 5: Thank you. A reference was added (line 65).

Minor. Typos. use of English

1. *Awkward sentences need to be edited on lines 53, 58, 69, 79, 121, 172, 205-206, 253, 268, 270, 297, 375, 377, 393-394, 431, 453, 520, 696-697*

Response: Thank you for pointing out the problem. These sentences were improved, and the manuscript was critically revised by two native English speakers (the language editing certificate was uploaded). For your convenience, we listed the new line numbers of related sentences here: Original line 53 (revised manuscript line 50-51), 58 (56-57), 69 (61-64), 79 (69-70), 121 (deleted), 172 (170-171), 205-206 (147-149), 253 (230-231), 268 (deleted), 270 (260-261), 297 (285-287), 375 (375-376), 377 (377-378), 393-394 (393-396), 431 (431-432), 453 (453), 520 (540-541), 696-697 (727-729).

2. *Line 83 change arrange to array*

Response: Thank you. The word 'arrange' was deleted, and related sentence was revised according another reviewer's suggestion (line 71-72).

3. *Line 120 add "The" before "majority"*

Response: Related sentence was deleted according another reviewer's suggestion.

4. *Line 197 change fusing to fusion*

Response: This issue was fixed (line 139).

5. *Line 186 reword. SGD1 does not have a new function*

Response: This sentence was deleted.

6. *Line 278 change bended to bent*

Response: We changed word (line 268).

7. *Line 380 change suffered to underwent*

Response: We modified it (line 381).

8. *Line 432-433 cite the E2 ligases reported to regulate grain yield.*

Response: We added a recent reference (line 433).

9. *Line 458 not clear what is meant by quality control*

Response: Thank you. A protein quality control system consisting of molecular chaperones and proteases controls protein folding and prevents the aggregation of misfolded proteins by either refolding or degrading aggregated species (Phillips et al.,

2020). This information and references were added to the manuscript to improve understanding (lines 485-488).

10. *Line 541 change trait to traits*

Response: We fixed it (line 560).

11. *Line 707 change grown to grew*

Response: Thank you, we corrected it (line 740).

Reference (for reviewer 3)

Chen, Q., Liu, R., Wu, Y., Wei, S., Wang, Q., Zheng, Y., Xia, R., Shang, X., Yu, F., Yang, X., Liu, L., Huang, X., Wang, Y., and Xie, Q. (2021). ERAD-related E2 and E3 enzymes modulate the drought response by regulating the stability of PIP2 aquaporins. *Plant Cell* 33, 2883-2898.

Lyu J, Wang D, Duan P, Liu Y, Huang K, Zeng D, Zhang L, Dong G, Li Y, Xu R, Zhang B, Huang X, Li N, Wang Y, Qian Q, Li Y. Control of Grain Size and Weight by the GSK2-LARGE1/OML4 Pathway in Rice. *Plant Cell*. 2020 Jun;32(6):1905-1918.

Phillips BP, Gomez-Navarro N, Miller EA. Protein quality control in the endoplasmic reticulum. *Curr Opin Cell Biol*. 2020 Aug;65:96-102. doi: 10.1016/j.ceb.2020.04.002.

Zhang, H., Zhou, J.F., Kan, Y., Shan, J.X., Ye, W.W., Dong, N.Q., Guo, T., Xiang, Y.H., Yang, Y.B., Li, Y.C., Zhao, H.Y., Yu, H.X., Lu, Z.Q., Guo, S.Q., Lei, J.J., Liao, B., Mu, X.R., Cao, Y.J., Yu, J.J., Lin, Y., and Lin, H.X. (2022). A genetic module at one locus in rice protects chloroplasts to enhance thermotolerance. *Science (New York, N.Y.)* 376, 1293-1300.

REVIEWERS' COMMENTS

Reviewer #1 (Remarks to the Author):

The manuscript is more clearly written now, although the Introduction and Discussion are still too long and discursive. The Editors should deal with this as the ms looks pretty long for Nature Comms.

The authors have done quite a lot more work and the results are now more solid and comprehensive. I have one concern about Figure 6h, showing the phosphorylation levels of BZR1 as an output of BRI1 signalling. In the results section the relationship between increased BL kinase levels and decreased BZR1 phosphorylation should be spelt out instead of the current vague text. At the moment this is spelt out in the Discussion, where it is too late for the reader to grasp the work. Also, is the phosphorylation of BZR1 assessed using phos-tag gels? It's unusual to see such large mobility shifts due to phosphorylation.

Reviewer #2 (Remarks to the Author):

Overall, a fantastic study, very thorough and rigorous- going from macroscopic phenotype to gene to biochemical function. It adds to an emerging story that the ubiquitin pathway impacts fundamental processes influencing agronomic traits. Not surprising result given the body of work in general that has emerged implicating the ub pathway is just about every process! Here the work is strong in demonstrating that the SGD1 gene in Setaria is responsible for the phenotypes. The significance of this effect is strengthened by creating mutants in wheat, maize and rice, demonstrating its conservation. The effect of loss of SGD1 in Setaria is supported by multiple mutants, and complementation with the coding region. Its biochemical function as an E3 ligase is confirmed.

Other strengths are identifying an interacting E2 and showing genetically its involvement in the same process. An impactful aspect is the allelic variation that can quantitatively influence the phenotype. Finally, the authors showed that SGD1 interacts with and ubiquitinates the brassinosteroid receptor, BRI1. The effect of this modification is not the canonical facilitation of degradation, but rather the protein is present at higher levels in presence of SGD1. In the revision, they extend the results by more directly showing changes in stability.

This work will be of interest to both basic and applied researchers. The authors also link this E2-E3 to BR signaling and propose that these proteins regulate cell size through regulating the BL receptor ubiquitination and in some way its function. Hormone biologists will be interested.

I greatly appreciate the authors' thoughtful and thorough responses to the previous reviews. The authors carefully considered the comments and responded with detailed reasoned answers. I think that the changes proposed by the reviewers improved the manuscript and I hope the authors feel that way because I appreciate how much work was involved to wade through three different reviewers' comments. Thankfully, some were the same amongst them! In the revised version, the data and analyses are strengthened. The data support the conclusions drawn and the writing is very clear. Methods are detailed.

I do not have any substantial concerns regarding this revision. I loathe to bring them up, but there are two sentences which I couldn't let by.

Lines 442-443. Remove the "can" in the sentence: "Moreover, we found that the E3 ligase SGD1 can directly interacted with and ubiquitinated the key BR receptor BRI" to "Moreover, we found that the E3 ligase SGD1 directly interacted with and ubiquitinated the key BR receptor BRI"

Lines 456-457. This sentence is a bit awkward: “However, it would be a challenge and an intriguing issue on figuring out how SGD1 keeps BR signaling at optimal efficiency through finetuning BRI1.” Suggest modifying to (but this is only a suggestion): However, it will likely be challenging to understand mechanistically how SGD1 keeps BR signaling at an optimal efficiency through the fine-tuning of BRI1 levels.

Reviewer #3 (Remarks to the Author):

The authors have satisfactorily addressed all of my comments and concerns in the revise manuscript.

Responses to Reviewers

Reviewer #1 (Remarks to the Author):

Comments

Comment 1: *The manuscript is more clearly written now, although the Introduction and Discussion are still too long and discursive. The Editors should deal with this as the ms looks pretty long for Nature Comms.*

Response 1: Thank you for your constructive and thorough review throughout the whole peer review process. These valuable comments and suggestions have helped to improve the coherence of our manuscript. As regard to comment 1, we have revised the manuscript and made changes to the Introduction and Discussion sections to make them more concise. Specifically, we have removed the descriptions on protein function of DA1, DA2, and GW2 from the introduction part as they are not closely related to our study. The related content was rewritten (lines 55-57). A few sentences that were indirectly connected to the topic in Discussion section were removed and related content was reorganized (lines 428-430, lines 454-457, lines 462, lines 467-470, lines 476-477, lines 482, lines 500-502, lines 507-511, lines 523-524, and lines 526-530). The introduction has been shortened to 612 words and the discussion has 1332 words. We compared this to a few published articles in Nature Communications and found that it meets the requirements. We hope these changes can address your concerns, and we are willing to make edits to the manuscript if you or the editor have any other requirements.

Comment 2: *The authors have done quite a lot more work and the results are now more solid and comprehensive. I have one concern about Figure 6h, showing the phosphorylation levels of BZR1 as an output of BRI1 signalling. In the results section the relationship between increased BL kinase levels and decreased BZR1 phosphorylation should be spelt out instead of the current vague text. At the moment this is spelt out in the Discussion, where it is too late for the reader to grasp the work.*

Response 2: We are pleased that you found our work more solid and comprehensive after revision. We apologize for any confusion caused by the vague text in Figure 6h. In the revised manuscript, we have spelled out the relationship between increased BL kinase levels and decreased BZR1 phosphorylation in the Results section (lines 326-333). We believe this will be clearer to readers now.

Comment 3: *Also, is the phosphorylation of BZR1 assessed using phos-tag gels? It's unusual to see such large mobility shifts due to phosphorylation.*

Response 3: Thank you. We understand your concern regarding the large mobility shifts of BZR1 due to phosphorylation. However, this is true for BZR1 protein partly because it has many phosphorylation sites. In Arabidopsis, there are 25 putative phosphorylation sites (Wang et al., 2002), while in foxtail millet there are 29 (Zhao et al., 2021). Out of these sites, 12 were verified by LC-MS/MS analysis (Tang et al., 2008) and point mutation experiments in Arabidopsis (Ryu et al., 2007). There is no need to use phos-tag gels, we

can clearly distinguish the phosphorylated and dephosphorylated forms of BZR1 using regular SDS-PAGE. A number of published papers have used the same method to analyze BZR1 phosphorylation, and the mobility shifts of BZR1 and pBZR1 bands are about 3-10 kDa in different experiments, which are similar to our result in Fig.6h (response Fig. 1A-D). Furthermore, our previous study has already demonstrated that the anti-SiBZR1 antibody can recognize SiBZR1 (Wang et al., 2021), and that BL treatment induces a change in SiBZR1 from its phosphorylated form to its dephosphorylated form (response Fig. 1E-F). The mobility shifts of BZR1 and pBZR1 bands are also similar to the results in the current study. Based on all of the above, we suggest that our analysis of the phosphorylated and dephosphorylated forms of BZR1 is reliable.

Response Fig. 1. The mobility shifts of BZR1 and pBZR1 bands in different publications. A. Detection of phosphorylated and unphosphorylated BZR1 in the absence (-OA) or presence of 250 nM okadaic acid (+OA) using BZR1::BZR1-CFP transgenic Arabidopsis seedlings (left, anti-YFP antibody) and homozygous *bin2-1* mutant (right, anti-BZR1 antibody) (Tang et al., 2011). B. Detection of phosphorylated and unphosphorylated BZR1 in 10-day-old transgenic Arabidopsis seedlings carrying proBZR1::BZR1-GFP or proBZR1::BZR12K/R-GFP using anti-GFP antibody (Srivastava et al., 2020). C. Immunoblots of BZR1-YFP using anti-GFP antibody in different Arabidopsis transgenic seedlings (Wang et al., 2021). D. OsBZR1 phosphorylation status in rice Nip and *wak11-2* samples exposed to eBL using anti-OsBZR1 antibody (Yue et al., 2022). E and F. Detection of SiBZR1 dephosphorylation induced by BL treatment in foxtail millet using anti-GFP antibody (Zhao et al., 2021) or anti-BZR1 antibody (Wang et al., 2021). All proteins in Response Fig.1 were separated by SDS-PAGE.

Reference

Ryu H, Kim K, Cho H, Park J, Choe S, Hwang I. Nucleocytoplasmic shuttling of BZR1 mediated by phosphorylation is essential in Arabidopsis brassinosteroid signaling. *Plant Cell*. 2007;19(9):2749-62.

Srivastava M, Srivastava AK, Orosa-Puente B, Campanaro A, Zhang C, Sadanandom A. SUMO Conjugation to BZR1 Enables Brassinosteroid Signaling to Integrate Environmental Cues to Shape Plant Growth. *Curr Biol*. 2020; 20;30(8):1410-1423.

Tang W, Deng Z, Oses-Prieto JA, Suzuki N, Zhu S, Zhang X, Burlingame AL, Wang ZY. Proteomics studies of brassinosteroid signal transduction using prefractionation and two-dimensional DIGE. *Mol Cell Proteomics*. 2008;7(4):728-38.

Tang W, Yuan M, Wang R, Yang Y, Wang C, Oses-Prieto JA, Kim TW, Zhou HW, Deng Z, Gampala SS, Gendron JM, Jonassen EM, Lillo C, DeLong A, Burlingame AL, Sun Y, Wang ZY. PP2A activates brassinosteroid-responsive gene expression and plant growth by dephosphorylating BZR1. *Nat Cell Biol.* 2011;13(2):124-31.

Wang C, Tang S, Zhang Q, Shang Z, Liu X, Diao X, Zhao M. Kinase activity is required for the receptor kinase DROOPY LEAF1 to control leaf droopiness. *Plant Signal Behav.* 2021;16(11):1976561.

Wang RJ, Wang RX, Liu M, Yuan W, Zhao Z, Liu X, Peng Y, Yang X, Sun Y, Tang W. Nucleocytoplasmic trafficking and turnover mechanisms of BRASSINAZOLE RESISTANT1 in *Arabidopsis thaliana*. *Proc Natl Acad Sci USA.* 2021; 17;118(33): e2101838118.

Wang ZY, Nakano T, Gendron J, He J, Chen M, Vafeados D, Yang Y, Fujioka S, Yoshida S, Asami T, Chory J. Nuclear-localized BZR1 mediates brassinosteroid-induced growth and feedback suppression of brassinosteroid biosynthesis. *Dev Cell.* 2002;2(4):505-13.

Yue ZL, Liu N, Deng ZP, Zhang Y, Wu ZM, Zhao JL, Sun Y, Wang ZY, Zhang SW. The receptor kinase OsWAK11 monitors cell wall pectin changes to fine-tune brassinosteroid signaling and regulate cell elongation in rice. *Curr Biol.* 2022 Jun 6;32(11):2454-2466.e7.

Zhao Z, Tang S, Li W, Yang X, Wang R, Diao X, Tang W. Overexpression of a BRASSINAZOLE RESISTANT 1 homolog attenuates drought tolerance by suppressing the expression of PLETHORA-LIKE 1 in *Setaria italica*. *Crop Journal.* 2021; 9(5): 1208-1213.

Reviewer #2 (Remarks to the Author):

General comments

Overall, a fantastic study, very thorough and rigorous- going from macroscopic phenotype to gene to biochemical function. It adds to an emerging story that the ubiquitin pathway impacts fundamental processes influencing agronomic traits. Not surprising result given the body of work in general that has emerged implicating the ub pathway is just about every process! Here the work is strong in demonstrating that the SGD1 gene in Setaria is responsible for the phenotypes. The significance of this effect is strengthened by creating mutants in wheat, maize and rice, demonstrating its conservation. The effect of loss of SGD1 in Setaria is supported by multiple mutants, and complementation with the coding region. Its biochemical function as an E3 ligase is confirmed.

Other strengths are identifying an interacting E2 and showing genetically its involvement in the same process. An impactful aspect is the allelic variation that can quantitatively influence the phenotype. Finally, the authors showed that SGD1 interacts with and ubiquitinates the brassinosteroid receptor, BRI1. The effect of this modification is not the canonical facilitation of degradation, but rather the protein is present at higher levels in presence of SGD1. In the revision, they extend the results by more directly showing changes in stability.

This work will be of interest to both basic and applied researchers. The authors also link this E2-E3 to BR signaling and propose that these proteins regulate cell size through regulating the BL receptor ubiquitination and in some way its function. Hormone biologists will be interested.

I greatly appreciate the authors' thoughtful and thorough responses to the previous reviews. The authors carefully considered the comments and responded with detailed reasoned answers. I think that the changes proposed by the reviewers improved the manuscript and I hope the authors feel that way because I appreciate how much work was involved to wade through three different reviewers' comments. Thankfully, some were the same amongst them! In the revised version, the data and analyses are strengthened. The data support the conclusions drawn and the writing is very clear. Methods are detailed.

Response to general comments

Thank you very much for your kind and encouraging comments. We are pleased that our work in *Setaria* satisfied the reviewers and may arouse interest among scientists in the field of plant science.

Of course, we sincerely appreciate the suggestions provided by you and the other two reviewers throughout this two-round peer review process. Your comments/advice have not only improved the current manuscript but also provided valuable direction for our future research. Thanks to your detailed guidance on scientific writing, the text is now more transparent and clear.

Specific comments

Comment 1: *I do not have any substantial concerns regarding this revision. I loathe to bring them up, but there are two sentences which I couldn't let by. Lines 442-443. Remove the "can" in the sentence: "Moreover, we found that the E3 ligase SGD1 can directly interacted with and ubiquitinated the key BR receptor BRI" to "Moreover, we found that the E3 ligase SGD1 directly interacted with and ubiquitinated the key BR receptor BRI"*

Response 1: We are grateful that the results of the requested experiments were to your satisfaction. Thank you for point out the problem in the sentence, we corrected it in our revised manuscript (line 431).

Comment 2: *Lines 456-457. This sentence is a bit awkward: "However, it would be a challenge and an intriguing issue on figuring out how SGD1 keeps BR signaling at optimal efficiency through finetuning BRI1." Suggest modifying to (but this is only a suggestion): "However, it will likely be challenging to understand mechanistically how SGD1 keeps BR signaling at an optimal efficiency through the fine-tuning of BRI1 levels."*

Response 2: Thank you for bringing the awkward sentence to our attention and providing a solution. We have revised it accordingly in our manuscript on lines 445-446.

Reviewer #3 (Remarks to the Author):

General comments

The authors have satisfactorily addressed all of my comments and concerns in the revise manuscript.

Response to general comments

Thank you very much for your positive and constructive review, which greatly improved our manuscript. We are pleased that our revisions were able to address all of your concerns and comments to your satisfaction.